# Metamaterials for Acoustic Noise Filtering and Energy Harvesting

**DOI:** 10.3390/s23094227

**Published:** 2023-04-23

**Authors:** Fariha Mir, Debdyuti Mandal, Sourav Banerjee

**Affiliations:** Integrated Material Assessment and Predictive Simulation Laboratory (i-MAPS), Department of Mechanical Engineering, University of South Carolina, Columbia, SC 29208, USA

**Keywords:** metamaterials, acoustic metamaterials, topological acoustics, noise barriers, energy harvesting, piezoelectric, piezoelectric energy harvesting, air ventilation, sound insulation, acoustic noise barriers

## Abstract

Artificial methods for noise filtering are required for the twenty-first century’s Factory vision 4.0. From various perspectives of physics, noise filtering capabilities could be addressed in multiple ways. In this article, the physics of noise control is first dissected into active and passive control mechanisms and then further different physics are categorized to visualize their respective physics, mechanism, and target of their respective applications. Beyond traditional passive approaches, the comparatively modern concept for sound isolation and acoustic noise filtering is based on artificial metamaterials. These new materials demonstrate unique interaction with acoustic wave propagation exploiting different physics, which is emphasized in this article. A few multi-functional metamaterials were reported to harvest energy while filtering the ambient noise simultaneously. It was found to be extremely useful for next-generation noise applications where simultaneously, green energy could be generated from the energy which is otherwise lost. In this article, both these concepts are brought under one umbrella to evaluate the applicability of the respective methods. An attempt has been made to create groundbreaking transformative and collaborative possibilities. Controlling of acoustic sources and active damping mechanisms are reported under an active mechanism. Whereas Helmholtz resonator, sound absorbing, spring-mass damping, and vibration absorbing approaches together with metamaterial approaches are reported under a passive mechanism. The possible application of metamaterials with ventilation while performing noise filtering is reported to be implemented for future Smart Cities.

## 1. Introduction

A worldwide spike in industrialization has led to an increasing amount of noise and noise pollution. Noise pollution is defined as the unwanted propagation of sound energy creating physiological problems. Airborne sound travels through the air. It can transmit, reflect, or be absorbed by obstructions. At times, it is difficult to track sound since it is invisible. As a result, the methods and instruments used to track and filter the sound are difficult to develop. Hence, constant research dollars are spent at various scales to tackle the sound. As a result, innovative and efficient materials are being designed as noise barriers to filter unwanted noise in various environments that reduce noise pollution.

Energy is a vital element in the growth of modern society. From a light bulb to outer space missions, we need energy everywhere. Some energy is visible to us, such as light, but most of the existing energies in nature are not visible. Among such energies, electrical energy is the most commonly used form. Due to the high demand for electricity, different measures are being utilized to convert other forms of energy to electricity. The process which derives energy from external sources is called Energy Harvesting. The converted or harvested energy can be stored in a capacitor or battery for later use. This stored energy can work as a power source for low-energy electronics. In recent years, phenomenal interest has been shown in the harvesting of ambient vibrational energy and converting it into electrical power. The motivation of the research is to power wireless remote sensors, which are usually powered by batteries. Batteries possess the disadvantage of having a finite lifespan which creates problems of frequent replacement, especially during emergencies. In certain situations, it is not feasible to replace batteries either because of the complicated structural situation or due to extra-sensitive circuits with the risk of damage and increased costs. Therefore, in these complicated situations, if ambient energy in the atmosphere can be utilized to harvest electric potential to power these batteries concurrently, then most of the existing problems could be solved. Various transduction mechanisms can be employed to harvest this kind of energy. One such mechanism involves the use of piezoelectric materials to harvest energy from the unused, trapped, or lost vibrational energy of the host structure.

Not only the ambient vibration but the sound and noise created by modern machinery and vehicles can also be used to harvest the energy by trapping them in a medium. In recent years road traffic has increased tremendously. It is, without a doubt, one of the most widespread sources of noise annoyance. Industries with heavy machinery introduce a significant amount of noise into the atmosphere. In such cases, acoustic noise barriers can be developed and incorporated to reduce the outbreak of such unwanted noises. However, the efficiency of these noise barriers has been a big question for a long time. Researchers are constantly working on improving the efficiency of the barriers, but to date, the maximum reported efficiency of noise barriers is around ~50%. Concerning traditional industrial barriers, new materials and designs are combined to create a more effective noise-filtering system. Sound-absorbing materials are being used to trap the sound inside the material and, thus, decrease the reflection of noise. The law of conservation of energy states that energy can neither be created nor destroyed—only converted from one form of energy to another. Hence, the energy that the sound barriers absorb is trapped inside the cells. This trapped energy is not being utilized anywhere. The trapped acoustic energy can be wisely harvested if certain measures or ways can be incorporated serving this purpose, resulting in ‘energy harvesting’.

Conventional materials do not show promising output for all these special applications. Therefore, metamaterials can be used for such applications. Metamaterials are engineered materials that possess unique properties that cannot be found in naturally occurring materials. Metamaterials can be constructed by assembling multiple materials or polymers. The properties of the metamaterial are different from the constitutional material properties, and they originate from the design assembly of the pieces. Acoustic metamaterials are metamaterials that are designed to control, direct, and manipulate sound waves, as these might arise in gases, liquids, and solids. The acoustic metamaterial follows the theory and outcome of negative index material. Since acoustic metamaterial is one of the branches of metamaterials, the basic principle of acoustic metamaterials is quite similar to the principle of fundamental metamaterials. These metamaterials usually derive their properties from the structure rather than the material composition, utilizing the inclusion of small inhomogeneities to enact effective macroscopic behavior. A negative refractive index of acoustic materials can be attained by manipulating and controlling the bulk modulus and mass density.

When we talk about using metamaterials to work as noise absorbers, we should consider what the safe level of sound is and how much sound needs to be filtered by the metamaterial in a noisy environment. The unit that measures the amount of sound is known as a decibel (dB). Decibel is the unit used to measure sound on a logarithmic scale. It is a logarithmic way of expressing a ratio. The logarithmic scale is used to scale down the range of audible sounds. The more decibels a sound has, the more forceful it is. Acoustic metamaterials absorb the impacted sound and minimize the sound decibels on the receiver end. The amount of sound intensity that decreases when passed through a structure is known as transmission loss. The higher the transmission loss is, the quieter the environment is on the receiver side. There is a universal integer rating that explains how well a noise barrier material minimizes sound. It is called Sound Transmission Class or STC. A Higher STC rating indicates better sound isolation. A standard material will have an STC between 20 s (such as glass) and 30 s (the average wall). Effective sound insulation appears at STCs around 50 s. STC testing standards are always updated, so an STC rating from two decades ago will not be the same as today. Any material with an average STC rating of 30 or higher is considered better for noise shielding. The STC rating is officially approved by the American Society of Testing and Measurement. However, this rating has a few shortcomings. One is that the rating does not consider frequencies lower than 125 Hz. Another is that this system is not utilized in other parts of the world. A material can have a high STC number but not block specific sounds such as rumbling traffic, reverberating construction, or the droning hubbub of office voices. Therefore, an alternative to STC is required that is acceptable throughout the world and which does not have major drawbacks. Rw is the best alternative to STC and is commonly used by most of the world. Rw is the Weighted Sound Reduction Index which is used to measure the sound insulation abilities of walls, floors, windows, or doors. It is an International Organization for Standardization (ISO) rating and part of the ISO 140 (acoustic) family. Rw ratings are like STC in that they follow familiar testing methods. However, they also differ quite a bit; for example, Rw covers a larger frequency range than STC. The Rw rating cannot be directly compared to an STC rating. Some professionals prefer Rw because it corresponds to the decibel scale. One can expect any material with an Rw rating of 50 to reduce the noise by 50 decibels corresponding to that specific material. Similarly, another number is used when constructing or choosing noise barrier material which is known as the Noise Reduction Coefficient, or NRC. It is a number that ranges between 0.0 and 1.0. This parameter indicates the average sound absorption performance of a material. If an NRC rating of a material is 0, that means the object reflects sound. If the rating is 1, it indicates that the material absorbs 100% of the impacted sound energy. In the real world, it is not possible to find material with an NRC rating of 1, i.e., 100% sound-absorbing material. 

If a metamaterial is designed in such a way that it absorbs most of the impacted sound, then it is possible that the absorbed sound energy is trapped inside the metamaterial. Since energy cannot be created or destroyed, it is quite possible to change the form of the energy and harvest it using piezoelectric material into electrical potential. Piezoelectric materials generate an electrical charge when mechanical stress is applied. Barium titanate, lead titanate, lead zirconate titanate (PZT), and polyvinylidene fluoride (PVDF) are some commonly used piezoelectric materials. Strategic placement of piezoelectric materials inside a noise barrier cell enables it to harvest the trapped or impacted energy. The process of obtaining electrical energy from other external sources, such as solar, sound, vibration, wind, kinetic energy, etc., is known as energy harvesting. 

Rapid developments in technology and industry have led to an increased amount of noise in the environment. Therefore, controlling unwanted interference is necessary to ensure the efficiency of people these days. Hence, researchers have developed various means of noise-controlling mechanisms. This chapter reviews the noise control mechanisms that have been developed so far for effective noise reduction in various atmospheres. This review paper aims to strategically review the existing noise-blocking techniques investigated by various researchers. It also categorizes the different structural designs used for energy harvesting purposes. Numerical formulas and equations are kept at a minimum here unless necessary for the explanation. This review is divided into two major parts:(a)Noise Control;(b)Energy harvesting.

## 2. Noise Control

### 2.1. Noise Control Mechanism

Because noise control has become an essential measure that needs to be taken in this current situation, this section reviews the various noise-controlling mechanisms used in industries these days. Noise is measured on a decibel scale. Figure 1 represents the relationship between the noise in decibels and sound pressure in pascals corresponding to different noise levels. To understand the noise level even better, real-life examples of certain sound levels are also added to the chart. A sound level below 85 dB is usually tolerable. A sound level above 85 dB may cause serious health hazards. In places with loud sounds above 85 dB, personal ear protection or some sort of noise barrier is required. The chart below also states the optimal noise levels which are considered safe. Additionally, it addresses the levels above which personal precautions are advised.

Considering the chart values above, it is very necessary to develop mechanisms to control noise and minimize it when it reaches above the optimal safety levels. Based on the sound control mechanism, the approaches can be divided into two sections: active and passive control. Active control uses electro-acoustical approaches to control noise. There can be two types of electro-acoustical approaches; one is to control the sound by creating an anti-phase signal using speakers when it emerges from the source. Another method is by suppressing the structural vibrations that are the primary cause of the sound. The passive control mechanism uses various methods to modify the environment around the source or the receiver. Passive control includes engineered materials that absorb, diffuse, or redirect the impacting sound. Both active and passive control mechanisms can be divided into sub-sections based on the physics they use for noise filtration. The flow chart shown in Figure 2 helps to visualize the classification of the noise control mechanism easily. This chapter categorically reviews and discusses the mechanisms shown in the chart in further detail.

#### 2.1.1. Active Control Mechanisms

Modifying and canceling sound field by electro-acoustical approaches is called active noise control. There are mainly two methods for active control mechanisms. First, utilizing the actuators as an acoustic source to produce completely out-of-phase signals to eliminate the disturbances. The second method involves the use of flexible and vibro-elastic materials to radiate a sound field, interfering with the disturbances and minimizing the overall noise intensity. A passive noise control mechanism is good for high-frequency noise sources; for noises with lower frequencies, active noise control mechanisms work better. This section reviews the active noise control mechanisms used for low-frequency noise control. A summary of active noise control is tabulated in Table 1.

##### Control of Acoustic Sources

Control of acoustic sources is also known as Active Noise Control (ANC). This concept uses three basic components—speaker, microphone, and controller, as reported by Elliott [1]. In this method, the noise created from a primary source is passed through a controller to generate an anti-phase signal. This anti-phase signal is transmitted by a secondary source in the controller. Here, the microphone is regarded as a primary source and the loudspeaker as the secondary source. This method works best in an enclosed environment. An increased number of primary and secondary sources can improve the noise reduction efficiency of the system. However, total noise cancellation is impossible due to various technical reasons such as phase lag and others. Elliott et al. also proposed that a feedforward controller can be installed to overcome the problems, but apparently, such systems face challenges during installation. 

The ANC has been of interest to researchers since the late 1980s, and they claimed that this method would transform future cabin noise control [2,3]. ANC is mostly utilized to reduce aircraft noise. In the past, ANC has been studied inside an actual aircraft cabin by numerous researchers [4,5,6]. Among all these, a maximum of 14 dB noise reduction was achieved. It was also reported that the efficiency of the ANC mechanism depends highly on the presence and distribution of secondary sources. Additionally, Elliott et al. found that at localized field points, their model could achieve up to 35 dB of noise reduction. Based on these basic experimental studies, Martin et al. numerically validated the results but added that there should be no more than 30 secondary sources in this scenario [7]. A few potential problems related to this method are that it increases the weight of the structure and works best if it is near the sound source. To solve this problem, Kestell et al. proposed adjusting the transmitted signal [8]. Though ANC is commonly used in aircraft, it has been recently considered for trains and automobiles, as well [9,10,11,12]. Lately, the development of ANC systems was summarized in a review article by Samarasinghe et al. [13]. This article describes various reasons why ANC has not yet been widely adopted by people. The reasons include the higher cost of implementation, the complexity of the hardware requirement, poor stability, etc. This is still a developing field, and researchers are working to overcome the drawbacks.

##### Active Damping Mechanism

From the discussion in the previous section, it is evident that ANC systems have various drawbacks. To overcome these flaws, a new system is proposed that not only controls cabin noise but also works toward structural vibrational reduction. The main difference between this mechanism is that it controls the noise indirectly, whereas the ANC controls noise directly. Sensing accelerometers are used to measure the vibration level, which is sent to the controller. Another set of accelerometers is placed near the control region to provide feedback and minimize the error. The entire process runs in a loop and, thus, reduces the sound developed from specific locations. This active damping mechanism is also known as Active Structural Acoustic Control (ASAC). 

Like the acoustic source control mechanisms, ASAC has also been studied rigorously since the late 1980s. The initial fundamental work on such systems was published by Fuller and Jones [14], where they achieved a cabin noise reduction of around 20 dB. This study was extended by a few other researchers considering an actual aircraft cabin [15,16]. This extended research had some loopholes because it did not consider the effect of the actuator position. Later this issue was addressed in a comprehensive study conducted on a rectangular enclosed space. This study suggested that the actuator location need not be limited to a specific location [17,18,19]. This study has made the ASAC systems capable of cabin noise control in aircraft. Constant study on this system has improved the mechanism by multiple folds. Since the ANC had a problem of being bulky because of the presence of a shaker attachment, Fuller et al. proposed replacing the shaker with a piezoelectric patch [20]. Many other researchers validated the concept later [16,21]. The use of piezoelectric patches revolutionized this system. These patches were lightweight, could undergo large strain deformation, had low power requirements, and were affordable. The only drawback with these piezoelectric patches was their brittleness. A huge amount of study has been conducted on ASAC since then [22,23,24,25] to better understand and improve the mechanism.

Though active noise control can be performed employing ANC or ASAC, very few studies have been conducted where both approaches have been deployed. Recently, Sas et al. used both ANC and ASAC in the cabin of a station wagon [26]. The study concluded that the ASAC provides a global stable cabin noise control compared to ANC. They suggested that both methods are suitable for local noise control, but ASAC is preferred for global noise control. As ASAC controls noise and minimizes structural vibration, it is in every way better than ANC. Still, research is being conducted to improve the methods by addressing the drawbacks.

**Table 1 sensors-23-04227-t001:** Summary of active noise control.

	Author	Suggestions/Strategies/Method
** *Control of acoustic sources* **	Elliott et al. [1]	Feedforward or feedback control using a reference signal, error signal, and secondary microscope.
	J. Tichy et al. [2]	Necessity of developing optimization techniques in secondary sources and controlling microphone locations.
	Silcox et al. [3]	Sound attenuation of ~15 dB observed inside a thin, elastic, cylindrical shell with fixed discrete monopole sources.
	Elliott et al. [5]	A ~13 dB sound reduction was measured during in-flight experiments using 16 loudspeakers and 32 microphones.
	Martin et al. [7]	Testing various aircraft models using a number of secondary sources to calculate sound attenuation.
	Kestell et al. [8]	Analytical model used to predict and compare the virtual sensor’s performance and experimental validation.
	Oh et al. [10]	A ~6 dBA sound pressure reduction inside an automobile using an active feedforward control system model.
	Botto et al. [11]	Fuzzy and neural modeling paradigms integrated into active noise reduction scheme to minimize noise in a railway coach.
	Liu et al. [12]	A ~4 dBA noise reduction in electric locomotive cab using proposed ANC system based on whole cab space.
** *Active damping mechanism* **	Fuller and Jones [14]	Structural–acoustic coupling between the shell and the field shows global attenuation of interior noise at resonant and forced vibration frequencies.
	Simpson et al. [15]	Effectiveness of active vibration control methods to minimize aircraft cabin noise.
	Mathur and Tran [16]	Experimental investigation results of sound minimization inside aircraft cabin using active structural acoustic control.
	Pan et al. [17]	Acoustic characteristics and sound absorption properties of different boundary conditions in a well-damped, enclosed rectangular space are measured and experimentally verified.
	Pan and Hansen [18]	Optimum location selection of a point force actuator to control sound through a panel with a cavity.
	Fuller et al. [20]	A 10–15 dB of interior noise control using piezoceramic actuators.
	Sun et al. [21]	Piezoelectric actuators are used to reduce both structural vibration and interior noise.
	Grewal et al. [23]	A ~28 dB noise reduction and ~16 dB vibration reduction for propeller-induced noise and vibration were achieved using ASAC.
	Palumbo et al. [24]	A control algorithm for ASAC in an airplane to control blade passage frequency using 21 actuators and 32 microphones.
	Niezrecki and Cudney [25]	PZT actuators are used to control fairing vibration and internal acoustic environment.
	Sas & Dehandschutter [26]	Adaptive feedforward control algorithm to reduce road noise inside a car cabin under different road conditions.

#### 2.1.2. Passive Control Mechanisms

Passive noise control refers to those methods that aim to suppress sound by modifying the environment close to the sound source. As no input power is required in these methods, passive noise control is often cheaper than active control. However, the performance of the passive system is limited to mid and high frequencies. Active control works well for low frequencies, and hence, the combination of two methods may be utilized for broader bandwidth noise reduction.

##### Helmholtz Resonator

Helmholtz resonators are referred to as fluid-filled (usually air) hollow containers with a narrow neck system. The Helmholtz resonator effect is caused by the motion of the air at its neck. The fluid inside acts as a spring, which helps the neck air to oscillate, thus, resulting in a spring-mass system. This phenomenon of air resonance in a cavity with a narrow neck is known as Helmholtz resonance. The Helmholtz resonator and its effects on acoustic media have been studied for a very long time, but around three decades ago, Fahy and Schofield [27] presented some disagreement with the existing designs. They experimentally proved the existence of two modes on either side of the resonator’s fundamental frequency. This study is considered a foundation for Helmholtz resonator design.

Helmholtz resonance is being utilized in passive noise control mechanisms by modifying their design parameters that enable the structure to dampen the sound. Numerous research endeavors have been undertaken to determine the most effective way to use adaptive Helmholtz resonators. Though Fahy and Schofield’s study is considered the foundation, Cummings later showed that their study lacked practicality since they used only a single resonator. Cummings proved analytically that the use of an array of resonators is required in cases where multiple frequencies of sound are involved [28]. The study of Cummings was later investigated further with advanced computational power tools [29]. From these studies, it could be concluded that a change in incident frequency can change the effectiveness of the resonator. Therefore, a unique resonator design is required to address the noises of various origins. Koopman and Neise [30] studied adjustable resonators to dampen centrifugal fan noise. Their experiment demonstrated a drop of about 29 dB of noise. However, they did not suggest any conclusive method. 

Helmholtz resonators have been used widely in the automobile industry to reduce cabin noises. As they might lower the aesthetic appeal of the automobile’s appearance, Franco proposed that the boot compartment could be used for such installations [31]. He suggested creating a slit between the boot and the passenger cabin. A few other researchers have also studied the installation of Helmholtz resonators to minimize cabin noise in automobiles [32,33,34,35,36]. Apart from automobile cabins, the Helmholtz resonator has also been used to minimize aircraft cabin noises. Initially, Laudien and Niesl [37] introduced the concept of adding perforation behind the seat of a helicopter and, thus, creating Helmholtz resonators without increasing the weight of the aircraft. Similarly, Helmholtz resonators have also been used as a noise barrier material to minimize industrial noises. Functionally graded Helmholtz resonators were used by Zhao et al. [38] in a bio-inspired spiral pattern for rainbow trapping (Figure 3). They utilized the passive noise-controlling feature of the Helmholtz resonators and arranged them in a space-saving design. Their design is suitable for places where weight and space are a constraint. They have numerically and experimentally proven that a spiral array of Helmholtz resonators can filter broadband acoustic waves.

On the other hand, Helmholtz resonance was utilized by Isozaki et al. on a planar notch filter with sub-wavelength thickness [39]. Their designs included multiple Helmholtz resonators (HRs) connected to a hole created in a plate (Figure 4). They arranged the HRs around the hole in different numbers and patterns, such as single, dual, triple, quad, pent, and hexagonal patterns. Their numerical and experimental results agreed with each other and helped them to conclude that increasing the number of uniformly arranged Helmholtz resonators decreased the transmission at the notch frequency.

Another interesting application of the Helmholtz resonator has been reported by Wang et al. [40], who developed a noise barrier for high-speed railways. They used a hexagonal configuration-based Helmholtz resonator for noise-filtering purposes. The report describes and compares two models of the same design but differs based on the presence of a noise collection input module. They also incorporated PVDF film (piezoelectric) inside the cavity to harvest power from the acoustic pressure. The study results indicate that the model with a noise collection input module works best in all circumstances, as the input module helps in magnifying the sound pressure. This design indicated promising results and, thus, can be utilized as a low-frequency noise barrier material. From the studies conducted by Wang et al. and Isozaki et al., it is evident that the change in the dimension of the resonator cavity and neck parameters can change the performance of an HR. As changing the neck parameters is easier compared to changing other resonator dimensions, various researchers have manipulated the neck material and design parameters of the HR to optimize the noise-controlling performance [41,42,43,44].

##### Sound Absorbing Material

Sound can cause double trouble if the barrier material reflects the incident sound, thus, creating more problems. Inspired by such cases, sound-absorbing materials have been developed for noise barriers that allow the barricade to absorb most of the incident sound and result in keeping the environment clean. Previously, membrane-type sound-absorbing materials were most popular for indoor and cabin noise mitigation [45,46,47]. Cai et al. reported an ultrathin coiling-based metamaterial panel that absorbs low-frequency sound with minimal thickness [48]. Figure 5a represents the design developed by Cai et al. On the other hand, Cavalieri et al. [49] reported a 3D multi-resonant design that has an average transmission loss of 16.8 dB Figure 5b. Parallelly, for low-frequency noise mitigation, double negative meta-structures were reported by Kumar et al. [50]. This study reported a 1D acoustic metastructure capable of exhibiting double negative parameters. This material can absorb an average transmission loss of 45 dB for frequencies below 500 Hz [50]. The flexibility of the design allows it to be used for lowering aircraft cabin noise. The design is represented in Figure 5c.

Another ultrathin metastructure was proposed by Li and Assouar, where a thin perforated plate with holes is placed on top of a rigid squared air cavity with a coiled chamber [52]. This structure can achieve perfect sound absorption at a frequency range of 125 Hz ([52] Figure 5e). A very classic design that has been used for ages is the Schroeder Diffuse (SD) mechanism. Zhu et al. used the SD mechanism and redesigned it in the form of an acoustic metasurface with ultrathin thickness [51]. This design has far-reaching implications in noise control ([51] Figure 5d). Various other researchers have also developed unique sound-absorbing metamaterials which help in reducing the sound in most noisy environments [53,54,55]. These materials have unique features and material properties that allow them to work as noise absorbers. This is still a developing field, and there is a huge prospect in this section in the upcoming times.

##### Acoustic Metamaterial

In the past two decades, huge progress has been made in the field of acoustic metamaterials. These materials have demonstrated tremendous potential for absorbing wider noise frequencies. Owing to the promising advancement, the designs developed by researchers so far can be used in practical settings for effective noise reduction in various environments. Acoustic metamaterials use a few basic physics to attenuate sound waves and filter noise. The section below discusses the five widely used physics for noise filtration.

##### Bragg Scattering

Let us consider two identical scatterers at a distance of ∆x. There is a time lag or phase difference between the incident and the radiated wave. The scattered waves interfere destructively with respect to each other. The distance between the scatterers leads to a strong destructive effect on wave propagation. This phenomenon is called ‘Bragg scattering’. Bragg scattering can be observed. It happens at the heart of the phononic crystal. In a crystal, at the frequencies where the Bragg condition meets, a frequency window is created through which no waves can propagate. Such a window is called a bandgap. Acoustic metamaterial uses Bragg scattering to create bandgaps so that the cells can be used as a noise barrier material. Figure 6a represents a tunable acoustic metamaterial with a square array of circular holes and resonators which is capable of filtering low-frequency noises. The numerical study from this model indicates that the locally resonant Bragg scattering bandgap can be controlled by controlling the amount of deformation [56]. On the other hand, Chen et al. [57] reported a triply periodic co-continuous acoustic metamaterial that can filter waves using the Bragg scattering phenomenon (Figure 6b). However, these metamaterials operate on a very high-frequency range, so it is not suitable for application in regular noise barrier walls.

Similarly, Casadei et al. [58] reported another tunable waveguide configuration where Bragg scattering bandgaps are combined with piezoelectric resonators to confine the propagation of elastic waves in a phononic crystal plate (Figure 6c). Additionally, numerous researchers have worked on developing unique acoustic metamaterial structures that used the physics of Bragg scattering [59,60]. Recently, Wen et al. proposed an acoustic metamaterial beam consisting of periodically variable cross-sections which combine the Bragg scattering and locally resonant bandgap mechanisms. This design resulted in reducing the vibration of the structure in a wide frequency range. Based on the numerical and experimental validation of the model, they concluded that bandgaps become stronger and wider with the increase in the sub-cell sizes [61].

##### Local Resonance

From the research conducted so far, it is evident that periodic structures have an influence on the propagation of waves. For instance, if we consider two oscillators, the coupling between them is strongest if they are degenerate. The coupling is expected to split the degenerate system of the wave and the local resonator onto the dispersion of waves. This effect is strongest at the point where two frequencies come across. Additionally, similar to the Bragg scattering effect, a window is present through which no wave propagation is possible. The frequency is not directed by the spacing of the periodic array; rather, the local oscillator’s frequency dominates it. The propagation of the waves can be modified in the vicinity of a specific frequency by coupling the wave to a local resonance with a similar frequency. This local resonance phenomenon is widely used for designing noise-filtering acoustic metamaterials. Manipulation of the design parameters can help achieve the desired bandgaps in a specific frequency range. Figure 7 shows some of the structures where researchers used local resonance phenomena for the filtration of sound. Krushyanska et al. exhibited an optimal design of locally resonant acoustic metamaterial [62]. They presented a detailed study on the influence of geometric and material parameters, filling fractions, and inclusion shape on the width of the lowest bandgap. They have advised using tungsten core material as an alternative to traditionally used lead or gold metals. This model creates bandgaps in a low-frequency range (Figure 7a). In 2000, Liu et al. [63] developed sonic crystals based on localized resonant structures that exhibited spectral gaps with a lattice constant two orders of a magnitude smaller than the wavelength. The study utilized composites having microstructures units consisting of a solid core material with relatively higher density and a coating of elastically soft material. The structure had centimeter-sized lead balls as the core material and was coated with a softer material (silicone layer) and combined and arranged to form an 8 × 8 × 8 simple cubicle structure. The local resonance mechanism was observed in the rigid core attached to the host medium via soft material. Over the years, numerous studies have been conducted on the interaction of surface waves with locally resonant materials limited to thin resonant layers or structures. Zeighami et al. [64] explored the opposite by distributing the resonators through the whole medium depth (thick layers) overlying a homogenous half-space. The study investigated the dynamics of Rayleigh-type surface waves propagating through a finite-thickness resonant layer. Their resonant layer comprises random distribution of discrete resonators in an elastic matrix. They applied an analytical framework to investigate the propagation of seismic waves through a deep barrier of resonators placed under soil, which was modeled as an equivalent resonant layer. Numerical methods and simulations were applied to verify their results and attenuation of the Rayleigh-type waves propagating across the resonant layer. Their study proved that the bandgap width is connected to the thickness of the resonant layer.

Muhammad and Lim [65] proposed a thin elastic plate where low-frequency bandgaps can be observed. This structure uses local resonance phenomena. Researchers recommend that the plate metamaterial can be used for sub-wavelength manipulation, including seismic shielding of civil infrastructures (Figure 7b). Lim also reported another dissipative multi-resonant pillared acoustic resonator that successfully amplifies the local resonance bandgaps [67]. The design of the unit cell is demonstrated in Figure 7d. These unit cells can be arranged periodically to generate wider bandgaps. This model efficiently works at higher frequency ranges. Ahmed et al. [68] reported a multi-cell metamaterial model with linearly varying core mass that utilizes the physics of local resonance phenomena. Mir et al. [66] developed a MetaWall with a concrete frame, rubber matrix, and lead resonator that demonstrates local resonance phenomena, exhibiting sound filtering and energy harvesting capabilities (Figure 7c).

##### Antisymmetric Deaf Band

Ao et al. reported a design of an acoustic metamaterial that can form a far-field image beyond the diffraction limit [69]. They studied a 2D array of coaxially layered rods in water. They observed transverse modes corresponding to zero effective density. These are labeled as deaf bands since they do not couple with normal incident longitudinal waves. Very recently, Indaleeb et al. [70] reported an acoustic metamaterial where a Dirac cone-like point is introduced. They presented a deaf band-based predictive model which has the potential to achieve an engineered Dirac cone. Their model includes PVC cylinder PnCs immersed in air. They experimentally validated the model to confirm the numerically generated orthogonal wave transport phenomena. 

Earlier, Indaleeb et al. [71] also reported a Dirac cone-like dispersion at the center of the Brillouin zone created due to accidental degeneracy. They observed that non-dispersive deaf band frequency remains unaltered from any arbitrary periodic structure made of PnCs. Their claims are validated numerically and experimentally, and they reported that orthogonal wave transport, negative refraction, and wave vortex exist at the deaf band-based engineered Dirac cone (Figure 8). Figure 8 Panel 1 (a–b) shows the stress mode shapes of the phononic crystal and the pressure mode shape of the surrounding air media for the top band (green), deaf band (red), and bottom band (blue) identified from the parent dispersion band structure presented in Panel 2 (a–f). These bands were identified because they tend to degenerate at the center of the Brillouin zone, i.e., Γ point. It was found that at the frequency of this degeneracy, owning to the antisymmetric mode shape of the deaf band, sound transmission drops close to zero, as depicted through the transmission coefficient plot across frequency in Figure 8 Panel 1 (c). The frequency zone of reduced transmission identified between the red dotted line in the figure has an inherent relationship with the dimension of the unit cell, the material properties, and the dimension of the phononic crystal. With specific frequency isolation in mind for Industry 4.0 application, the above understanding could help create new acoustic noise barriers by design. It is to be noted that this phenomenon is primarily due to the degeneracy at the center of the Brillouin zone, which is unlike the degeneracies at the Brillouin boundary discussed in the following section.

##### Acoustic Quantum Hall Effect

The acoustic quantum Hall effect is a new phenomenon observed in acoustic metamaterials recently. Under a strong magnetic field, many intriguing phenomena can be observed. Recently, the introduction of graphene has opened the door to quantum transport control by mechanical means. Wen et al. [72] reported the first experimental realization of a giant uniform pseudo-magnetic field in acoustics by introducing a simple uniaxial deformation to acoustic graphene. They proposed a strategy to create a uniform pseudo-magnetic field (PMF) for airborne sound by arranging a 2D sonic crystal array in a triangular lattice and validated the model experimentally. They attained uniform PMF by modifying only one geometric parameter in a single direction. They claim that their model can be extended to other artificial structures, such as a patterned photonic crystal slab on a silicon chip (Figure 9).

Please note that this phenomenon is predominantly at the K point or at the Brillouin zone boundary.

Zhou et al. [73] developed a membrane-type metamaterial (MAM) and observed the quantum Hall effect. To theoretically analyze the metamaterial, they employed a membrane-based model with heterogeneous density. However, the working frequency range of the topological system is very narrow. The frequency range can be broadened by applying an electrical voltage (Figure 10). Figure 10a shows the architecture or arrangement of the phononic crystals in the proposed MAM. Figure 10b shows the dispersion curve obtained from the unit cell arrangement and the formation of the Dirac cone at the Brillouin boundary (i.e., K point). The experimental results have validated the concept of metamaterial with tunable topological behavior. It is lightweight and can be coupled with acoustic waves. The capacity of guiding the currents towards a specific direction according to the spin of the traveling electrons has been transposed to the classical domain in the field of electromagnetics and acoustics, unveiling the pseudo-spin locking of the guided waves. These macroscopic photonic/phononic crystal analogs are intrinsically wavelength-scaled. Following a similar concept, Yves et al. [74] reported an acoustic analog of the valley–Hall effect in the audible regime using a lattice of soda cans (Figure 11). They experimentally demonstrated the unidirectional excitation of the sound, guided at a scale much smaller than the wavelength of operation. These results not only opened a new direction for tantalizing valley–topological phenomena to the audible regime but also allowed us to envision compact applications for acoustic manipulation.

This is a recently developed phenomenon in the field of acoustics. Researchers are constantly working to understand its effects and how to manipulate it for more user-friendly operations. Studies conducted so far have shown a promising future for crystalline metamaterials for discovering more solid-state physics phenomena.

##### Topological Effect

In electrodynamics, topological insulators have drawn significant attention due to their unique one-way wave propagation characteristics, which are not affected by defects or disorders in the structures [75,76,77]. In topological insulators, the outer surface is conductive while there is a bandgap-like phenomenon inside the cell which makes it an insulator to the electron flow. Topological metamaterials are developed considering the same physics, and they demonstrate a new vision for the domination of wave propagation aside from Bragg scattering and local resonance. Understanding topological transition by the interaction between these mechanisms is strongly desired to extend the degrees of freedom in the design for this intriguing wave phenomenon. Quantum Spin Hall Effect (QSHE) is one such topological behavior that could be used for acoustic isolation and/or predictive back scattering immune one-way wave propagation along the domain wall. 

Lee and Iizuka [78] demonstrated a phononic metamaterial consisting of C-shaped elements and investigated the interaction between Bragg scattering and local resonance. They reported that by adding resonance scattering, a topological bandgap is opened from a Bragg scattering-based Dirac cone, and its bandgap is controlled by the resonance frequency of the cavities relative to the Dirac cone frequency. They proved that topological bandgap opening induced by the Bragg scattering could be reversed into an ordinary state or vice-versa by the thoughtful inclusion of local resonance (Figure 12). This situation is formed due to the degeneracies of double Dirac cones at the center of the Brillouin zone. At the ordinary state (when the local resonators are away from the center of the unit cell with a higher geometric diameter (Figure 12a,b), the mode shape of the lower bands with di-polar modes are the p-state and the mode shape of the upper bands with quadrupolar mode are the d-state, created a bandgap in between them. However, when the local resonators are towards the center of the unit cell with reduced diameter, the dipolar and quadrupolar states flip. If a metamaterial is created having these two states of the material keeping a domain wall between them, the acoustic wave will guide through this boundary, keeping the rest of the material insulated. Higher-order topological insulators are a family of recently predicted topological phases of matter that obey an extended topological bulk–boundary correspondence principle. Xue et al. [79] presented an acoustic metamaterial-based second-order topological insulator. They reported that the model is shape-dependent, and it allows the corner states to act topologically protected but reconfigurable local resonances (Figure 13).

Recently, a new class of topological states has been reported, which is localized in more than one dimension of a D-dimensional system [80] (see Figure 13 shows for Kagome lattice). Such systems are referred to as higher-order topological (HOT) states in 3D. These systems offer an even more versatile platform to confine and control classical radiation and mechanical motion. The assembled 3D topological metamaterial represents the acoustic analog of a pyrochlore lattice made of interconnected molecules and is shown to exhibit topological bulk polarization, leading to the emergence of the boundary states (Figure 14).

##### Spring Mass Damping System

The spring-mass-damper model is the most basic method of suppressing vibration. These models are used widely used to minimize the structural vibration and, thus, reduce the noise. This is the age of metamaterials, and therefore, the traditional spring mass damper system has also entered the metamaterials club for enhanced structural vibration suppression. Peng et al. [81] modeled a metamaterial plate with mass-spring damper sub-systems (Figure 15a). This design uses the local resonance phenomena of the sub-system to absorb energy and, thus, create stopbands. A bandgap was observed right above the natural frequency of the system. This model can be used to suppress low-frequency vibrations and expand the stopband. He, Xiao, and Li [82] presented a laminate acoustic metamaterial that has a carbon fiber-reinforced polymer and a periodic array of spring-mass damper sub-systems (Figure 15b).

This model acts as a vibration absorber, and the dispersion analysis shows that the model generates a wide stopband. The model was utilized to design a vehicle door where the vibration of the door was suppressed significantly. Various other researchers have also reported unique designs for such noise control mechanisms [83,84].

Colquitt et al. [85] numerically and analytically performed a study on arrays of resonators attached to the elastic substrate, which can be conceived as the spring-mass system involving the interaction between Rayleigh-type waves and sub-wavelength resonators. The analytical results displayed that the flexural resonances to the resonators coupled weakly into the substrate with the main effect incoming from the compressional modes of the resonators and, thus, opening the bandgaps. Palermo et al. [86] utilized a similar concept where their study proves it analytically and numerically but also performed experimentally to further support this system. Their study involved developing a metabarrier as a shield from seismic surface waves. In their experiment, a polymer resin block represented an artificial soil in which they embedded small-scale resonators over 12 lines in a triangular fashion. The resonator structure primarily involved a rigid aluminum shell, a heavy steel mass, and a soft spring. The experimental results were successful in attenuating the Rayleigh waves when passed through the metastructures. In addition, Yilmaz et al. [87] demonstrated a study on an alternative method that utilized effective inertia of wave propagation medium amplification using embedded amplification mechanisms. The study results displayed inertial amplification producing wider bandgaps at lower frequencies.

##### Vibration Absorbing Structure

Vibration is one of the major sources of industrial noise. Reducing the vibration of a structure or isolating the vibrating surface will lead to a huge reduction in environmental noise. Ideally, springs are used in most cases to suppress the vibrations. Modern complex problems with vibrational sources have led researchers to develop lattice structures to absorb a wide range of vibrations. Such vibration-absorbing structures fall under the passive noise-controlling mechanism. Yu et al. [88] reported a viscoelastic damping system that damps vibration for mid-range frequency noise control. Their model significantly reduced the total acoustic energy of the system and ensured a uniform modal energy distribution (Figure 16a). Feng et al. [89] reported a design to reduce the vibration of an existing glass window using a viscoelastic material. Their method enables retrofitting existing glass windows for vibration absorption (Figure 16b).

These windows work on a very low-frequency range (0–50 Hz). Their model was easy to implement, almost maintenance-free, and had aesthetic appeal. Sound transmission and insulation of the aeronautical panels represent one of the major problems in aircraft comfort. Valvano et al. [90] proposed a finite plate element with an advanced higher-order kinematic field for the analysis of noise reduction using a passive control system in the laminated structures. The acoustic insulation of the panels was evaluated by computing its sound transmission factor using the Rayleigh integral method. They conducted various numerical investigations to validate and demonstrate the accuracy and efficiency of the acoustic optimization procedure for the design of the viscoelastic plates (Figure 16c).

Table 2, Table 3 and Table 4 summarizes all the research and models discussed so far.

### 2.2. Acoustic Metamaterial with Ventilation

In recent years there have been huge studies conducted on acoustic metamaterials and their use as a noise-filtering material. However, for most of the designs, there is a prevailing complaint, which is blocked airflow. For an enclosed space, it is required that the circulation of fresh air from the external environment is brought into the room. For this reason, an acoustic metamaterial design with ventilation has been developed. Researchers have developed unique designs of metamaterial-based windows that allow airflow, are lightweight, highly tunable, and can be 3D printed. 

Recently, Kim and Lee presented a soundproof transparent window design with air ventilation [92]. The window design has a negative bulk modulus, and the reduction in sound level is recorded between 20 and 35 dB in the frequency range of 400–5000 Hz. They studied six different designs of the cell to compare the best effective model. The design developed by Kim and Lee is demonstrated in Figure 17.

Various other researchers have developed noise-filtering material designs that aid airflow. A few of the recent most effective designs can be seen in Figure 18 [83,84,85,86,87,88,89,90,91,92,93,94,95,96,97,98]. These studies indicate that sound transmission can be minimized even using barriers with air ventilation capacity. These models are an indication that soon the noise barrier industry is going to be revolutionized by a new design trend.

Over the years, various researchers have been developing several unique design structures for simultaneous noise cancellation and proper air ventilation. Recently, acoustic metamaterials have been explored and categorized into six major divisions for efficient noise filtration and air ventilation. They are based on acoustic facade systems, Helmholtz resonators, coiled-up space structures, metacage structures, and acoustic meta-absorbers. This section will primarily discuss the mentioned designs, their fundamentals, and their effect on air ventilation and simultaneous noise filtration.

#### 2.2.1. Acoustic Facade Systems

Skyscrapers are very common in big cities, and there are more every day. It has been observed that these skyscrapers are often installed with transparent double-leaf facade systems as their window panels. Researchers have been exploring these facade systems and their structure and materials for effective noise reduction and air ventilation. Bajraktari et al. [99] utilized an acoustic structure with two-faced sheets (primary and secondary facades) along with ventilated openings. The unique structural design and the openings resulted in the circulation of sound and air within the cavity between the structures before the secondary passage. The design caused frictional resistance due to the ventilated openings, thus, resulting in impedance mismatching from the air cavity enabling significant noise reduction and ventilation. This structure displayed a noise reduction of approximately 27–35 dB. Similarly, Martello et al. [100] customized the double-facade system by adding a thin layer of polyurethane conglomerate (sound absorbent) beneath the lightweight metal louvers. This design showed a significant reduction in noise with proper ventilation. Researchers have also explored the design of plenum window panels on double-facade window systems for sound transmission losses and air ventilation. The plenum window has the design of two-faced sheets but with staggered openings for the inlet and outlet passage. The presence of the zig-zag configuration leads to effective blockage of the sound path, leading to transmission losses without interrupting the effective air ventilation [101]. Many researchers investigated and worked on these plenum window structures to modify and enhance the transmission losses and effective ventilation. Lee et al. [102] investigated enhancing the efficacy of the plenum windows design by incorporating sonic crystals into them. The sonic crystals resulted in significant noise reduction up to 4.2 dBA and 2.1 dBA at a resonant frequency of 1000 Hz. The sonic crystals installation, although proven to be effective in noise reduction, possesses the disadvantage of requiring a larger space for installation and, at times, reducing the aesthetic feature of the structures. Figure 19 below describes the different acoustic facade systems for potential noise reduction and air ventilation.

#### 2.2.2. Helmholtz Resonators (HRs)-Based Acoustic Structures for Airflow

The next type of simultaneous noise reduction and air ventilation structure involves the use of Helmholtz resonators. Many researchers have been widely investigating the incorporation of HRs in their design for noise filtration. Kim, Lee, et al. [92] established a prototype of air-transparent soundproof window panels by integrating the 3D arrays of diffraction-type HRs consisting of a central hole having a diameter of subwavelength. The HR contributed to negative bulk modulus and the central hole leading to noise reduction and efficient air ventilation. Likewise, Kumar et al. [103] investigated an acoustic metamaterial-based window panel system that consisted of a square-shaped hole, ventilated hole, and HR with two square-shaped on the inner side walls of the system. The design was proven to substantially reduce the noise in low-to-mid frequency ranges, allowing better air ventilation simultaneously. Similarly, Wang et al. [104] presented a plate-type acoustic metamaterial structure consisting of a perforated plate and support frame for air ventilation and noise filtration. The design is composed of an array of multiple unit cells, with each unit cell comprising an orifice-like structure at the center. The thin perforated plate was mounted on the frame as the final structure. The design revealed the ability to produce transmission losses up to 25 dB at 430 Hz, enabling sound cancellation and sufficient air ventilation. Figure 20 describes the design of the different HR-based acoustic metamaterials for effective noise cancellation and supportive air ventilation.

#### 2.2.3. Acoustic Metacage Systems

Another interesting design often explored by researchers for effective noise reduction and air ventilation is the acoustic metacage system. These systems mostly comprise acoustic metamaterials installed to the confined sound source in the form of cage-shaped systems. The metacage is usually introduced to the sound source system and, at times, also to the receiver based on the requirement and compatibility. Melinkov et al. [105] recently established the feasibility of noise reduction and air ventilation by designing and installing a metacapsule/cage prototype to stage machinery. The design comprised multiple arrays of equidistant C-shaped meta-atoms in the three sidewalls of the cuboid design. The sound-absorbing pads were installed on the two other sides of the cuboid, followed by the rigid frame on the last face of it. The meta-atoms in the cage were equivalent to the HRs, thus, enabling noise reduction, whereas the air gap between the meta-atoms allowed the uninterrupted flow of air. Figure 21 demonstrates an acoustic metacage system.

#### 2.2.4. Acoustic Meta-Absorber Systems

Acoustic meta-absorbers are metamaterials that are capable of high absorption of sound in the targeted frequency range. The phenomena are mostly achieved through the concept of impedance matching, which emphasizes the coupling of incident acoustic energy with the absorbers. Basically, the acoustic impedance complements the air leading to null reflection from the sound absorber metastructure enabling maximum absorption of sound. Lee et al. [106] demonstrated an acoustic metastructure having a structure of dual resonance unit cells. Each particular cell in the structure consisted of two resonators: one with a hollow cavity at the neck and the other filled partially with foam. Based on the operating resonant frequency, the particular dimensions were designed, and it resulted in high absorption of noise at 900 Hz. Another example of acoustic meta-absorbers was demonstrated by Kumar and Lee et al. [107], where acoustic structures were designed for broadband sound absorption and effective ventilation. Their unique design consisted of orchestrated six labyrinthine-type unit cells in a hexagonal fashion. The subwavelength dimensions of the structure were able to achieve an effective sound absorption to a broad range of 400–1400 Hz. The design also consisted of a central opening that permitted unhindered air ventilation. Figure 22 below shows the schematic designs of a few acoustic meta-structures for sound absorption and air ventilation simultaneously.

#### 2.2.5. Acoustic Coiled-Up Space Metastructure Systems

Another set of acoustic metastructure involves subwavelength dimensions of space-coiled structures, at times in helical, labyrinthine, curved, and folded shapes. These structures are quite effective in reducing sound, as when the sound propagates through these structures, the longer coiled path and the frictional air resistance offered by the structure enable the modulation of the sound waves and reduction of the sound speed, thus, lowering the net sound intensity. Sun et al. [108] demonstrated a coiled-up-shaped metastructure composed of a central hollowed-out orifice and a peripheral helix pathway with varied pitch. The metastructure consisting of helical pathways demonstrated flexibility for tailoring the sound waves and resulted in high noise cancellation of about 90%, ranging from 900 to 1418 Hz. The central hollowed orifice structure is credited for sufficient airflow, thus, providing ventilation. Likewise, many other researchers have worked on this type of structure for effective noise cancellation and airflow. Figure 23 shows the schematic of a coiled-up space acoustic metastructure for noise cancellation and air ventilation.

## 3. Energy Harvesting

### 3.1. Energy Harvesting Based on Sources

Since energy is neither created nor destroyed, the sound and vibration energy that we are filtering using passive control mechanisms can be harvested in the form of electrical energy. Piezoelectric materials can be used to scavenge this unused energy in the form of electrical power. In this section, piezoelectric energy harvesting based on vibration and sound sources can be reviewed categorically. Piezoelectric materials in the form of solid crystals are used for tapping high-frequency energy sources, while polymer-based piezoelectric membranes can be used for low-frequency energy sources. The tree diagram in Figure 24 represents the classification of piezoelectric-based energy harvesting and its sources.

#### 3.1.1. Vibration Sources

Based on the vibrational sources, energy harvesting approaches can be classified into two major categories, intermittent and continuous. The continuous source represents the models where the host structure vibrates at specific frequencies or a band of frequencies, such as machine vibration. The intermittent source does not rely on the input frequency; however, the host structure deforms and generates power upon the availability of the source, such as footsteps. One of the major differences between continuous and intermittent sources is their operating principle. While the resonance phenomenon is the key to generating maximum power using the continuous source, the intermittent source uses pure bending mode to harvest energy. 

Puscasu et al. [109] introduced a new technology for converting energy generated from steps into electricity. Their model was built from the available piezoelectric membranes arranged in a rectangular array. This model was able to generate 17.7 mJ of useful electrical energy per activation to provide up to 10.6 s of light (Figure 25).

Harvesting energy from vibrations is the most common form of energy harvesting. Cantilever structures are widely utilized in most of these cases. A few researchers have used spiral and twisted structures as well. Lueke et al. [110] presented a folded spring-like structure for vibration-based energy harvesting. Two classes of folded spring energy harvesters were developed. This design was capable of harvesting energy within frequencies ranging from 45 to 3667 Hz. This model harvested a maximum of 69.5 nW at 226.3 Hz. Liu et al. [111] developed an S-shaped MEMS PZT cantilever that could harvest energy from a frequency as low as 30 Hz. A few other researchers have also worked on spiral-shaped cantilever energy harvesters [112,113,114,115].

In the case of vibration-based energy harvesting, galloping piezoelectric energy harvesters are used in places where there are chances of induced vibration from the airflow. In these models, there is an extra mass attached to the tip of the beam. Self-excited vibration is induced when the tip-bluffed body is subjected to airflow. Various researchers, including Ewere and Wang [116], have investigated the performance of galloping energy harvesters [117,118,119,120]. Figure 26 represents a schematic of a galloping piezoelectric energy harvester.

Sun et al. proposed a U-shaped vibrational-based piezoelectric energy harvester (U-VPEH) [121]. They investigated the properties of the model analytically and validated the results experimentally. The report states the comparison of the linear and non-linear models side by side, and it was found that the non-linear U-shaped model narrows the bandgap. The maximum voltage response changes during the up-sweeping and the down-sweeping signals. Voltage responses of 8.743 V at 6.5 Hz under the up-sweeping and 14.18 V at 15.41 Hz under the down-sweeping signals were recorded. The experimental results demonstrated that the voltage response and the resonance frequency of the U-VPEH agree with the analytical and theoretical analysis. The design is represented in Figure 27.

The use of unimorph and bimorph energy harvesters is also a developing part of vibrational-based energy harvesting. Depending on the host structure and purpose of the harvester, damping is often required for the optimum power output. A unimorph is a cantilever-based structure that has one active layer, while a bimorph has two active layers. Bimorphs can have passive layers in between two active layers. Unimorphs and bimorphs have proven to be the most promising method for microscale energy harvesting [122,123,124,125,126,127].

Cottone et al. [128] introduced a non-linear buckled beam-like structure that is studied under the wideband on random vibrations. The design comprises a thin steel beam with two bonded layers of piezoelectric material (Figure 28). Numerical and experimental results are in harmony, and they indicate that such a structure has the capability of an extremely high-power generation when in the buckled state compared to the unbuckled state.

#### 3.1.2. Sound Sources

Generally, acoustic energy is ultimately dissipated into thermal energy at the propagation stage, and the low- and mid-frequency sound waves have attracted the most attention. One of the reasons is that the specified frequency band of noise is usually a significant component of the spectrum. The other reason is that at the low- and mid-frequency range, the corresponding sound wavelengths are longer, making it quite difficult to absorb or isolate them using most engineering structures. Many approaches have been developed to effectively absorb or isolate low- to mid-frequency acoustic sound waves; these include both passive and active approaches. The mechanism of sound energy harvesting can be explained easily in Figure 29. The ambient noise passes through either a resonator or an acoustic metamaterial, where it gets trapped. The trapped energy is then converted into electric potential by using either a piezoelectric, electromagnetic, or triboelectric energy conversion mechanism. This energy can be stored in batteries for further application.

Acoustic energy harvesters may be small in size, but they can harvest a significant amount of power output. A combination of metamaterials and membrane-type piezoelectric materials is widely used for acoustic energy harvesting these days. Li et al. reported a membrane-type energy harvester that is also capable of low-frequency sound insulation [130]. The basic structure consists of a circular TPU membrane fixed by a rigid aluminum ring. A rigid aluminum mass is attached on both sides of the membrane. A flexible PVDF membrane (piezoelectric) is used for energy harvesting (Figure 30a). Results show that power output is observed in the nano-Watt scale, which is comparatively low. Wang et al. [131] proposed a compact acoustic energy harvesting system comprising a beam-based PZT transducer and a dual layer of an acoustic metamaterial. To effectively improve the efficiency of the cell, the first bending mode of the transducer is designed for the resonating vibration related to the amplification effect. The authors report that under 100 dB incident pressure, the maximum output voltage is calculated to be 72.6 mV, which is almost 4.2 times higher than the models without the layered metamaterial (Figure 30b). An innovative and practical acoustic metamaterial energy harvester was proposed by Qi et al. [132], where a defect was created on the plate to accommodate the PZT patch. Using this model, a maximum output voltage of 1.3 V and a power density of 0.54 $\mu W/{\mathrm{cm}}^{3}$ were obtained at a frequency of 2257.5 Hz (Figure 30c).

Helical designs have been in fashion for various purposes for quite a long time. Yuan et al. [134] presented a helical structure for low-frequency acoustic energy harvesting that can be 3D printed (Figure 30e). A PZT patch is used in the design for energy harvesting, which gives a power output of 7.3 µW at an acoustic resonance frequency of 175 Hz and to a 100 dB sound pressure level excitation. A few other researchers have also developed helical structures for acoustic energy harvesting. Yuan et al. reported a metallic substrate with proof mass that harvests energy in the mW range [133]. Sonic crystals and Helmholtz resonators both have been used for waveguiding and energy harvesting for a very long time. Yang et al. incorporated the two basic structures into one module for harvesting acoustic energy [136]. Experimental results show that the proposed harvester exhibits ~23 and ~262 times higher maximum harvesting efficiencies than the sonic crystal resonator and the Helmholtz resonator structure, respectively (Figure 30g). 

Very recently, Mir et al. [137] proposed an acoustic energy-harvesting spiral multifunctional material that includes cylindrical Helmholtz resonators with various heights and diameters arranged in a spiral form. The inclusion of the piezoelectric membrane in the model makes it capable of harvesting energy from acoustic pressure while filtering noise simultaneously. The design of this model can be observed in Figure 31.

Table 5 summarizes the energy harvesting modules discussed here.

## 4. Future Recommendations for Multifunctional Designs

Because the world is progressing, more cost-effective and efficient noise barriers are becoming more necessary these days. In this review, various designs of noise barriers have been listed and compared. Although most of them have very interesting designs and promising outputs, they are not always feasible for real-life applications. Apart from having increased manufacturing and installation costs, most of them are not capable of surviving real-life environments. Additionally, most noise filtering designs are not equipped with energy harvesting capability. Hence, either the existing designs can be modified to be more cost-effective by including energy harvesters, or novel materials can be designed and manufactured to incorporate the desired properties. Figure 32 demonstrates the ultimate choice of the design that is envisioned to be developed in the future.

Another demanding concern regarding noise barriers is the issue with airflow. Though most of the designs are very effective in noise filtering, air flow blockage is a concern for most of them. Recently, many researchers have proposed various designs with enhanced air ventilation facilities. Based on the proposals presented by various researchers in this article, it is evident that such shortcomings can be resolved soon. 

Overall, a general directive should be developed that includes the discoveries reported so far and the shortcomings that are expected to be resolved.

## 5. Conclusions

This paper presents a detailed review of acoustic noise filtering and simultaneous energy harvesting mechanisms. The entire study is divided into major sections and sub-sections for ease of understanding. A few conclusions can be drawn based on the present review of three overlapping technologies and futuristic requirements. Future requirements may include a) a material system or structure that is able to filter acoustic waves or noise within the audible range (20 Hz–20 kHz) below ~65 dB; b) a system for acoustic noise filtering that, preferably, does not block airflow and/or visibility (please note that blocking visibility and airflow have been the traditional approach); c) noise barriers that produce some form of renewable energy from ambient vibration and acoustic noise because, with increasing energy demand, they will be more applicable for the future with a higher return on investment compared to standing idle, which is the present day scenario. Based on the above three driving requirements, a review is conducted, and state-of-the-art technologies are reported in this article. All technologies reported herein are sufficiently mature. Some may need more research and investment than others. Based on the domain of application, appropriate physics of noise filtering could be adopted, as reported in this article. Nevertheless, a goal-driven research roadmap could be created based on this study. Some very specific conclusions are enumerated below.

Though various models have been developed for noise-controlling purposes, and the structures have shown promising output in laboratory experiments, many of them have complexities associated with them which make them hard to transfer and apply to a real-world scenario. Similarly, the metamaterials with energy harvesting ability have comparatively low-power harvesting output. There is a huge scope of research that can be performed and investigated in this domain to increase the power output from such smart materials. Additionally, one of the major concerns with traditional noise barriers is blocked airflow, and it is very impressive to learn that many researchers have worked successfully on developing noise filters with ventilation for fluid flow. So far, these have only been used for acoustic noise filters, but we believe that advanced research in the future can incorporate smart materials into such systems to harvest energy from the structures simultaneously.

## Figures and Tables

**Figure 1 sensors-23-04227-f001:**
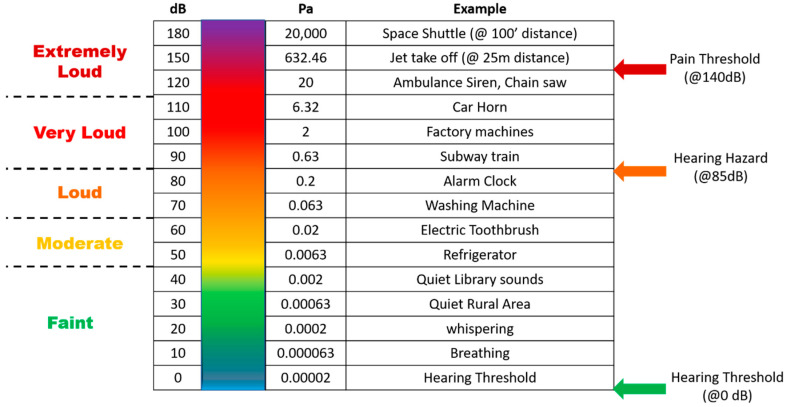
Comparison of noise level with daily life examples.

**Figure 2 sensors-23-04227-f002:**
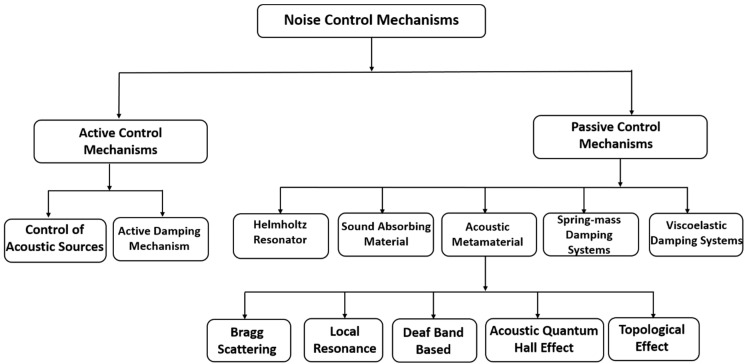
Classification of noise control mechanism.

**Figure 3 sensors-23-04227-f003:**
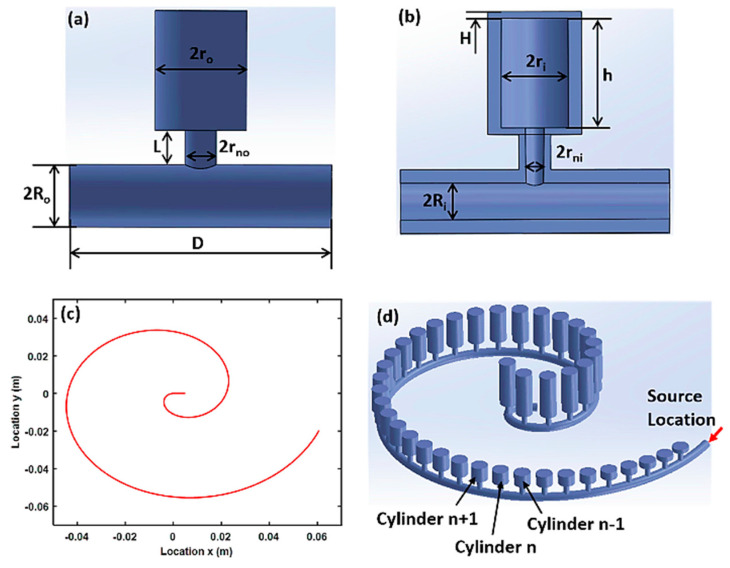
Schematic illustration of the cochlear-inspired structure. (**a**) Front view of Helmholtz resonator unit cell. (**b**) Cross-sectional view of the Helmholtz resonator unit cell showing the hollow interior of the cylinder and spiral tube. (**c**) Archimedean spiral and (**d**) isometric view of the entire structure [38].

**Figure 4 sensors-23-04227-f004:**
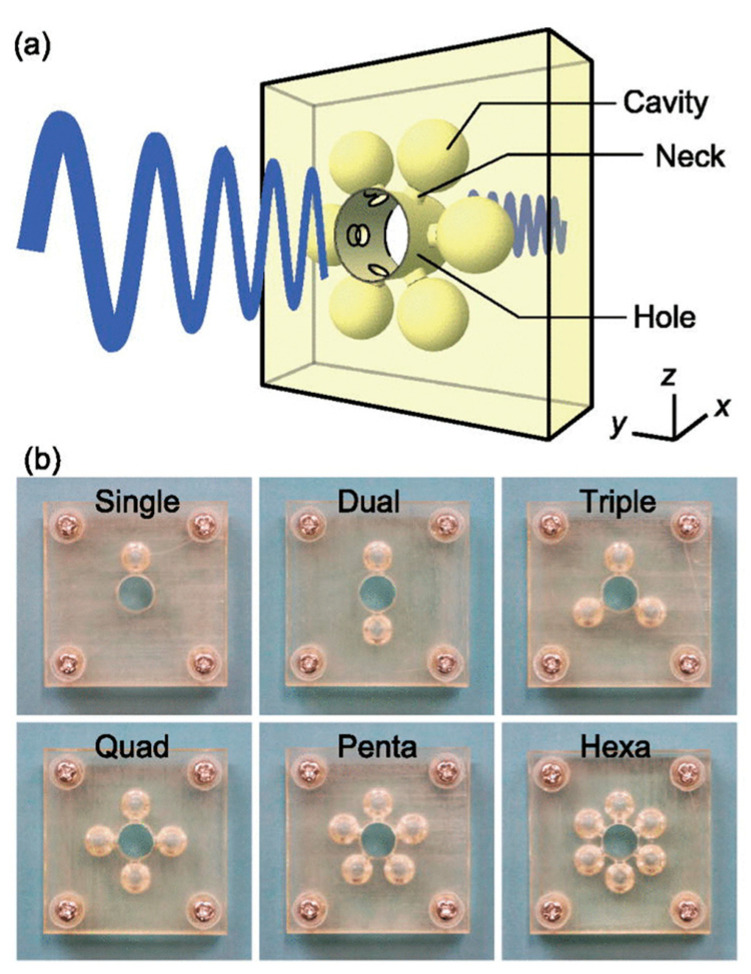
(**a**) Schematic illustration of the proposed planar acoustic filter. (**b**) Photographs of the fabricated filters [39].

**Figure 5 sensors-23-04227-f005:**
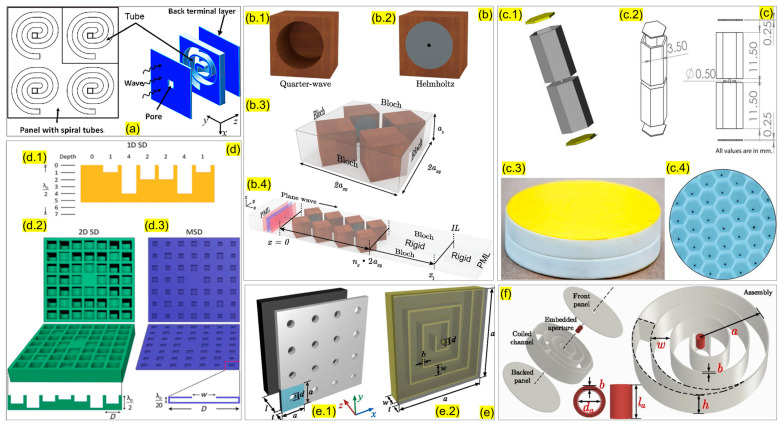
Sound-absorbing materials developed by various researchers. (**a**) Ultrathin coiling-based metamaterial panel [48]; (**b**) 3D multi-resonant sound absorbing metamaterial: (**b.1**) Quarter-wavelength, (**b.2**) Helmholtz resonator, (**b.3**) Three dimensional unit cell for eigenvalue problems, (**b.4**) Scattering problem for the calculation of the IL of an infinite LRSC slab [49]; (**c**) 1D metastructure with double negative parameters: (**c.1**) schematics of the unit cell, (**c.2**) Dimensions of the cell (**c.3**) Photograph of the fabricated metastructure (**c.4**) Top view of the metastructure without membrane [50]; (**d**) Ultrathin acoustic metasurface-based Schroeder diffuser (**d.1**) 1D Schroeder diffuser, (d.2) 2D Schroeder diffuser (**d.3**) proposed metasurface based Schroeder diffuser [51]; (**e**) ultrathin metastructure with (**e.1**) thin perforated plate with holes is placed on top of (**e.2**) a rigid squared air cavity with a coiled chamber [52]; (**f**) Acoustic perfect absorbers via spiral metasurface with embedded apertures [53].

**Figure 6 sensors-23-04227-f006:**
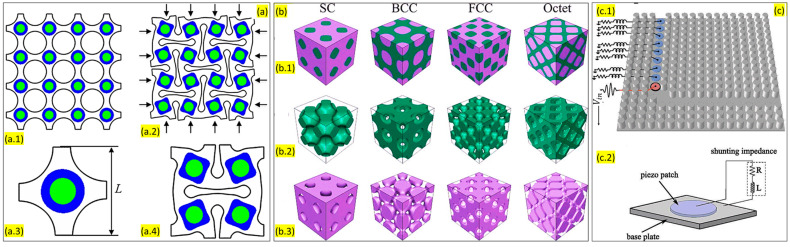
Bragg scattering phenomena observed in various acoustic metamaterial structures. (**a**) Tunable acoustic metamaterial with (**a.1**) square array of circular holes and resonators (**a.2**) geometry of (**a.1**) can be recognized by instability subjected to equibiaxial compression (**a.3**) primitive RVE in undeformed configuration, (**a.4**) enlarged RVE in deformed configuration [56]; (**b**) Triply periodic co-continuous acoustic metamaterial capable of filtering waves using the Bragg scattering phenomenon (**b.1**) 2 × 2 × 2 unit cells with simple cubic lattice, body centered cubic lattice, face centered cubic lattice, face centered cubic lattice and octet-truss lattice, (**b.2**) corresponding phase A in these metamaterials, (**b.3**) corresponding phase B in these metamaterials [57]; (**c**) Piezoelectric resonator arrays for tunable acoustic waveguides and metamaterials (**c.1**) schematic diagram of the phononic crystal plate with cylindrical stubs and L shape waveguide, (**c.2**) schematic diagram of the unit cell of the vertical channel [58].

**Figure 7 sensors-23-04227-f007:**
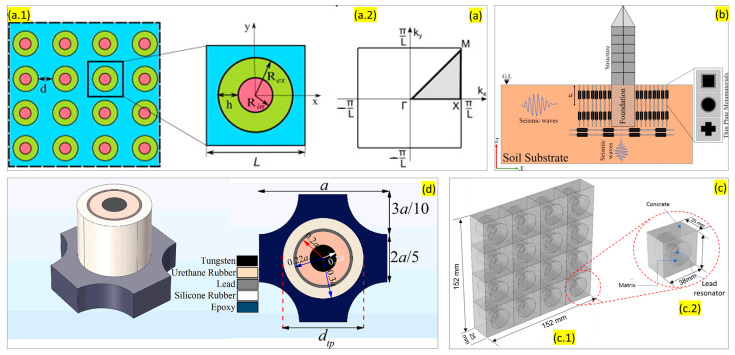
Local resonance phenomena observed in various structures. (**a**) Locally resonant metamaterial: (**a.1**) Optimal design cross section of the metamaterial and corresponding unit cell, (**a.2**) first Brillouin zone (square) and irreducible Brillouin zone (triangle) for a square lattice; [62]; (**b**) Elastic wave propagation in thin wave metamaterial [65]; (**c**) Design demonstration of MetaWall noise barrier (**c.1**) entire brick, (**c.2**) unit cell [66]; (**d**) Energy harvesting using sub-wavelength scale acousto-elastic metamaterial [67].

**Figure 8 sensors-23-04227-f008:**
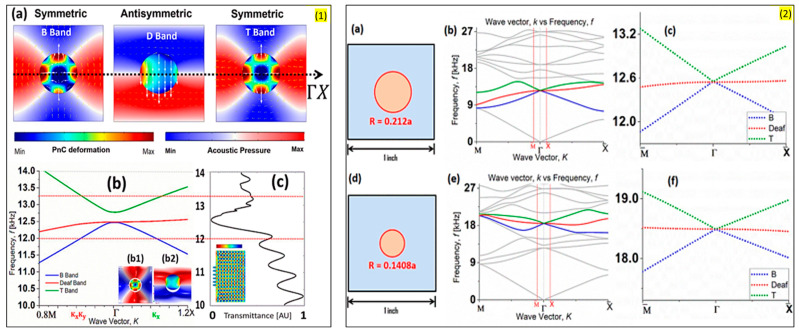
(**1**) (**a**) B, deaf, and T band mode shapes of the PVC cylinder surrounded by the air pressure mode shapes with arrows, (**b**) band structure before tuning, identified near region A, (**c**) numerical calculation of the transmit showing almost zero transmission near deaf band; (**2**) Accidental degeneracy for region A and B: (**a**) Unit cell for region A with PnCs of radius r = 0.212a in air matrix (**b**) Dispersion relation for region A (**c**) magnified view of the Dirac like point for region A, (**d**) Unit cell for region B with PnCs of radius r = 0.1408a in air matrix, (**e**) dispersion relation after decreasing the radius, (**f**) magnified view of the Dirac like point for region B [71].

**Figure 9 sensors-23-04227-f009:**
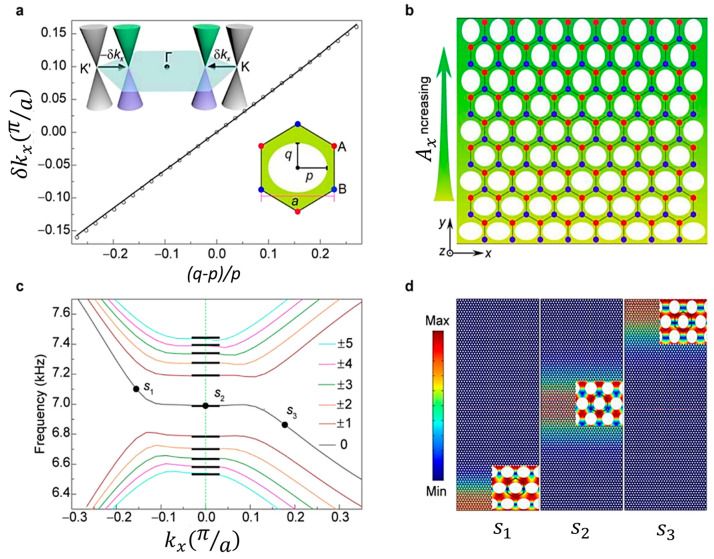
Synthesized acoustic magnetic field and relativistic Landau quantization for observing acoustic quantum Hall effect (**a**) simulated Dirac point shifts in momentum space, (**b**) Landau gauge potential sonic crystal with a linearly varying ξ along y direction and a transitional invariance along x direction, (**c**) spectrum near the Dirac frequency, (**d**)pressure amplitude distribution for three eigenstates labeled in (**c**) [72].

**Figure 10 sensors-23-04227-f010:**
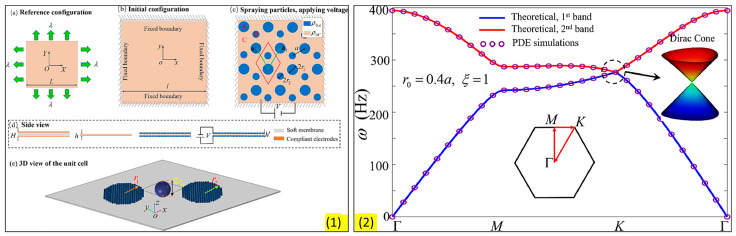
(**1**) Fabrication steps of soft MAM (**a**) reference configuration, (**b**) Initial configuration, (**c**) spraying particles and applying voltage, (**d**) side view of the cell (**e**) 3D view of the unit cell. (**2**) The band structure and the Dirac cone in soft MAM [73].

**Figure 11 sensors-23-04227-f011:**
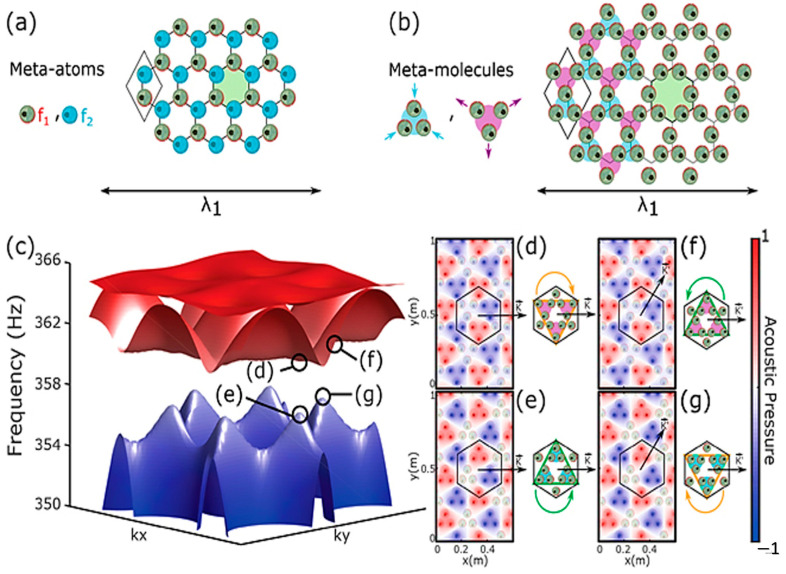
Visualization of (**a**) Bidisperse honeycomb lattice, (**b**) breathing Kagome lattice of soda cans. (**c**) Dispersion relation of the breathing Kagome lattice of cans and (**d**–**g**) acoustic field map of the crystalline mode at the valley K and K’ [74].

**Figure 12 sensors-23-04227-f012:**
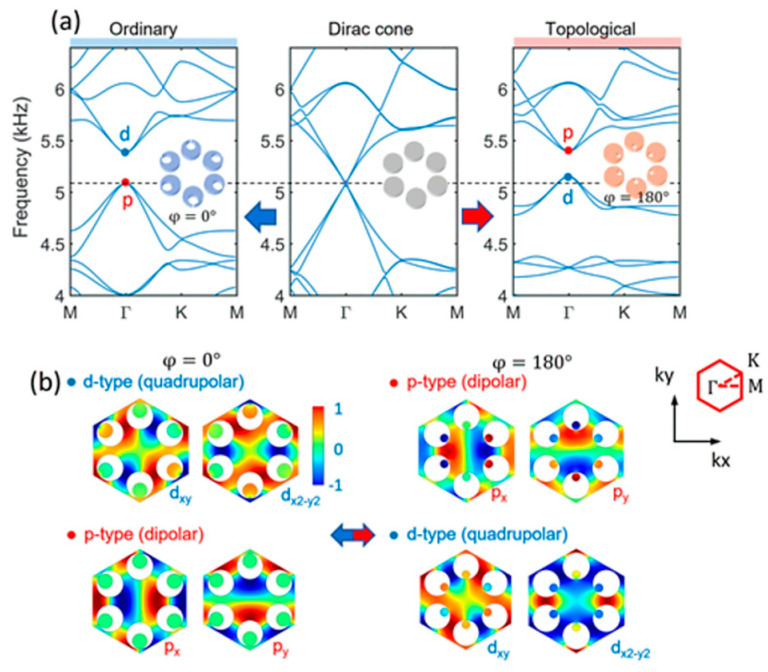
(**a**) Band structures of the honeycomb phononic crystals for the ordinary state (left), double Dirac cone (middle), and topological state (right). (**b**) Pressure fields for the ordinary (left panel) and topological (right panel) states [78].

**Figure 13 sensors-23-04227-f013:**
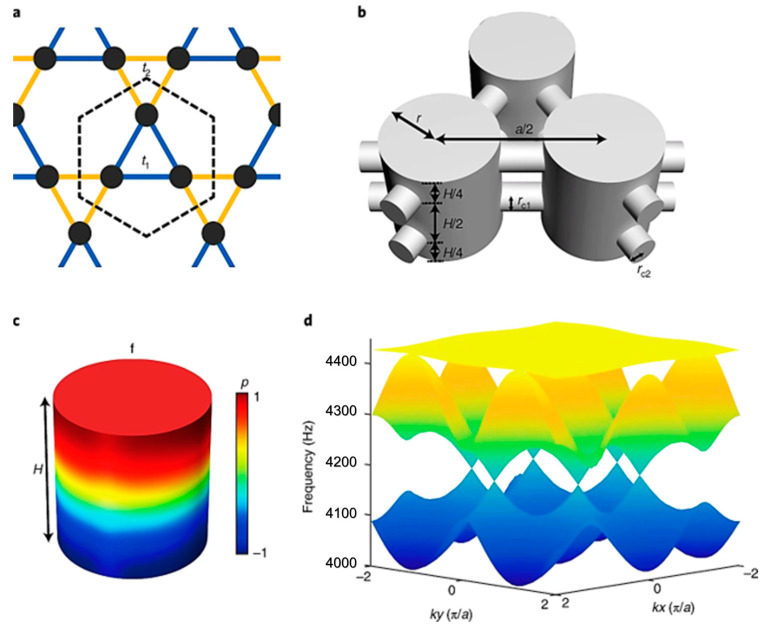
(**a**) Tight-binding model for the kagome lattice; (**b**) Unit cell of the acoustic kagome lattice, with a cylindrical resonator at each site joined by thin waveguides; (**c**) Topological mode shape; (**d**) Numerically computed bulk bands for the acoustic kagome lattice [79].

**Figure 14 sensors-23-04227-f014:**
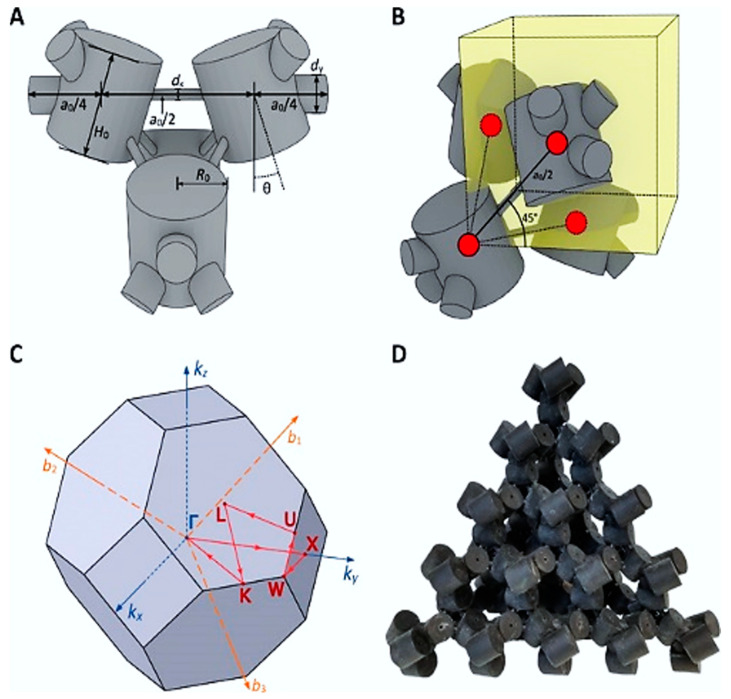
(**A**) Schematic and (**B**) realistic design of the Wigner-Seitz unit cell of the expanded pyrochlore lattice (**C**) first Brillouin zone of the FCC lattice (**D**) photograph of the 3D topological metamaterial assembled from 3D printed metamolecules, with boundary cells attached [80].

**Figure 15 sensors-23-04227-f015:**
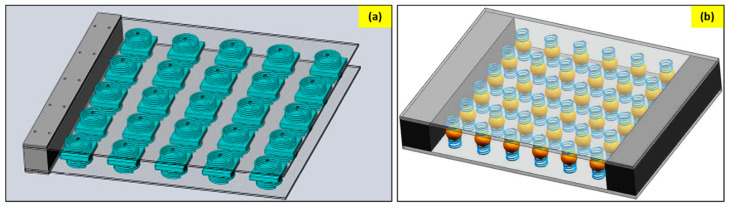
Spring-mass damping systems to suppress noise. (**a**) Acoustic metamaterial plate with elastic wave absorption [81]; (**b**) Structural vibration suppression in laminate acoustic metamaterial [82].

**Figure 16 sensors-23-04227-f016:**
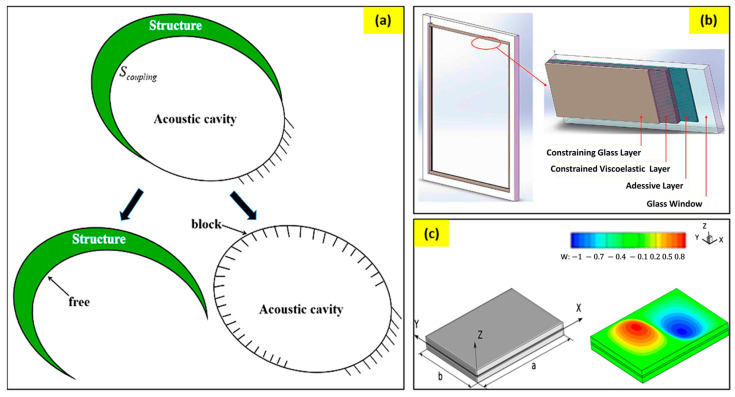
Vibration absorbing structure (**a**) viscoelastic damping for mid-frequency noise control [88], (**b**) vibration reduction of an existing glass window [89], (**c**) noise reduction passive control system based on viscoelastic material-based retrofit [90].

**Figure 17 sensors-23-04227-f017:**
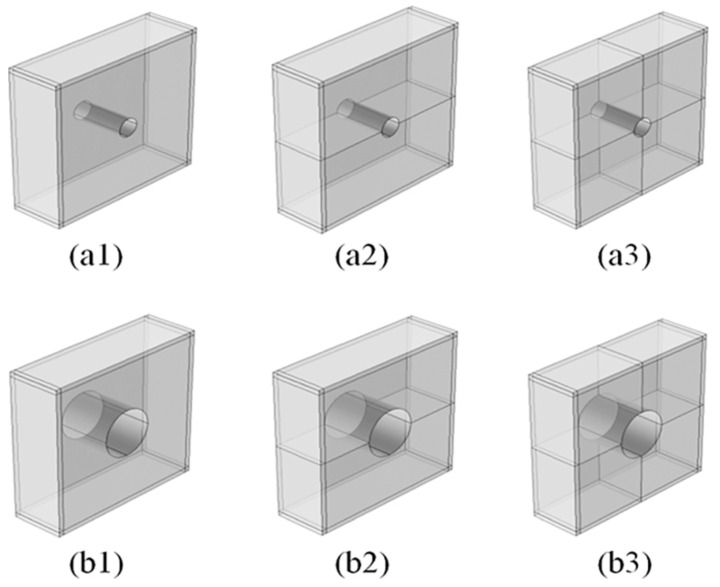
Diffraction resonators or acoustic cells. Diameters of the air holes: 20 mm for (**a1**–**a3**), and 50 mm for (**b1**–**b3**). There are three structures: one room for (**a1**,**b1**), two rooms for (**a2**,**b2**), and four rooms for (**a3**,**b3**) [92].

**Figure 18 sensors-23-04227-f018:**
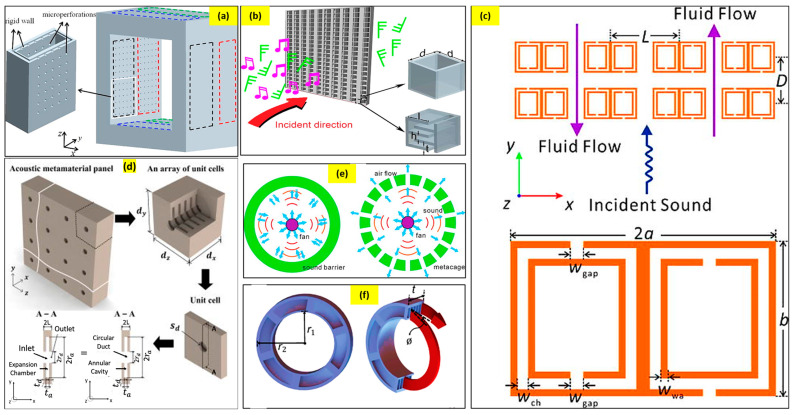
Acoustic metamaterials with ventilation for fluid flow. (**a**) Broadband acoustic absorber with ventilation performance [95], (**b**) omnidirectional ventilated acoustic barrier [96], (**c**) high efficiency ventilated metamaterial at low frequency [93], (**d**) acoustic metamaterial for fluid passage and soundproofing [98], (**e**) acoustic metacages with steady air flow [94], and (**f**) ultra open metamaterial silencer [97].

**Figure 19 sensors-23-04227-f019:**
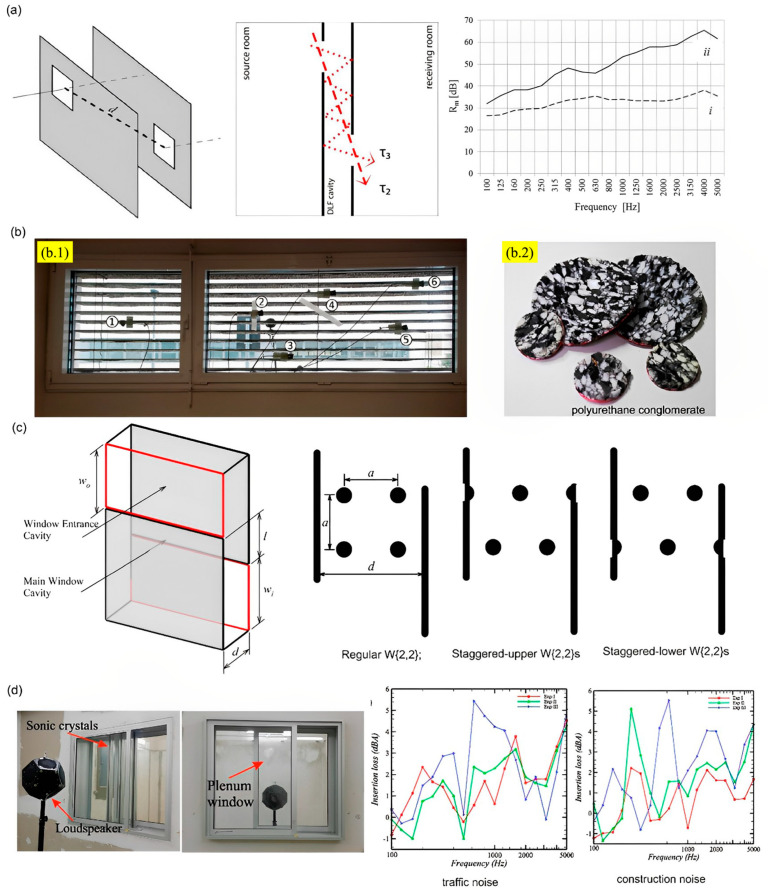
Acoustic metamaterials with ventilation for airflow and noise reduction using (**a**) double-facade system with a plot of measured sound reduction index (Rm) of the experimental wall (i: one layer fully closed and one fully open; ii: both layers fully closed) [99], (**b.1**) Photographs of the measurement set-up of the installed glazed surface (1–6) and (**b.2**) the absorptive material (polyurethane conglomerate) used with the louvers [100], (**c**) zig-zag staggered window configuration [101], and (**d**) sonic crystals-based window system and the plots having different settings and the measured insertion loss spectra under traffic noise an [102].

**Figure 20 sensors-23-04227-f020:**
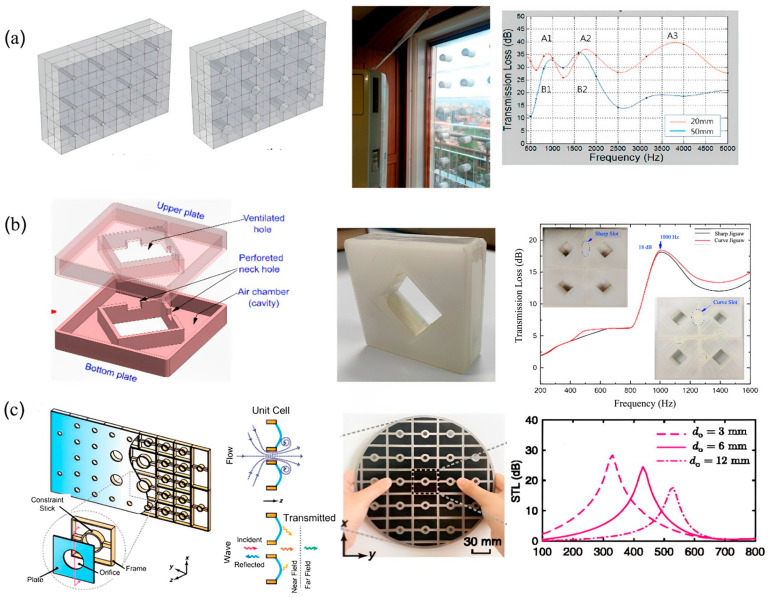
Helmholtz resonators-based acoustic metamaterials with ventilation for airflow and noise reduction using (**a**) HR-based transparent window panel [92], (**b**) HR-based ventilated metamaterial unit cell composed of two necks and a single chamber [103], (**c**) HR-based perforated and constrained acoustic metamaterial [104].

**Figure 21 sensors-23-04227-f021:**
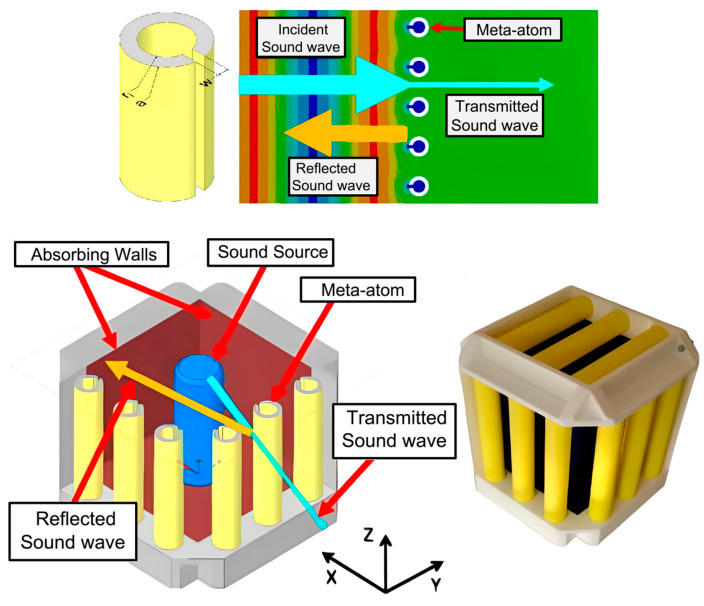
Acoustic metacage with ventilation for airflow and noise reduction having C-shaped meta-atoms and metacage capsule [105].

**Figure 22 sensors-23-04227-f022:**
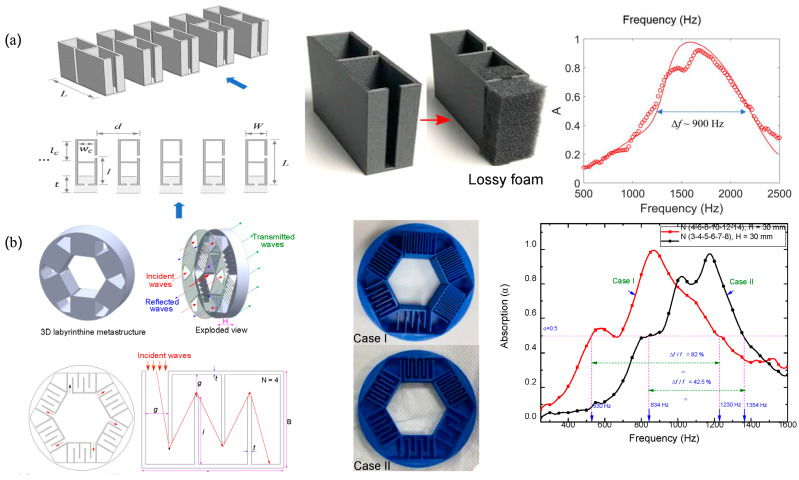
Acoustic meta-absorbers with ventilation for airflow and noise reduction using (**a**) dual resonators-based cell unit [106] and (**b**) hexagonal orchestrated six labyrinthine type unit cells [107].

**Figure 23 sensors-23-04227-f023:**
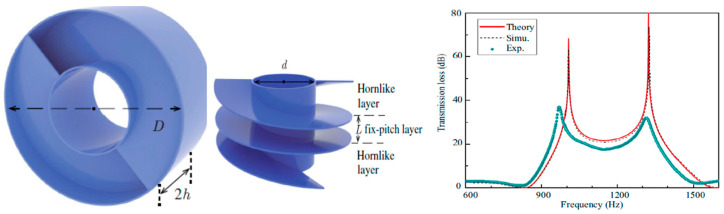
Coiled-up space acoustic metamaterial structure having a central orifice for air ventilation and a coiled-up helical pathway for noise reduction [108].

**Figure 24 sensors-23-04227-f024:**
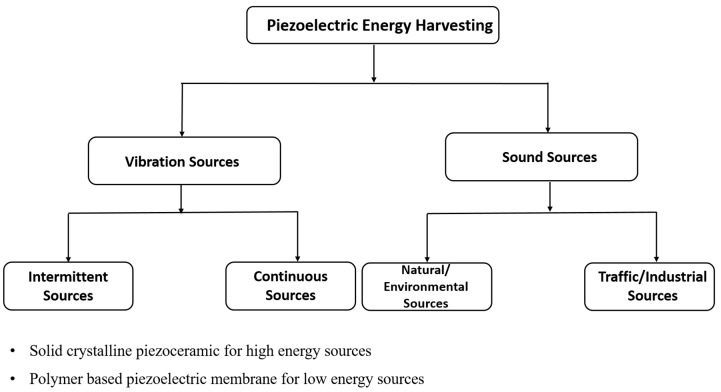
Classification of piezoelectric energy harvesting sources.

**Figure 25 sensors-23-04227-f025:**
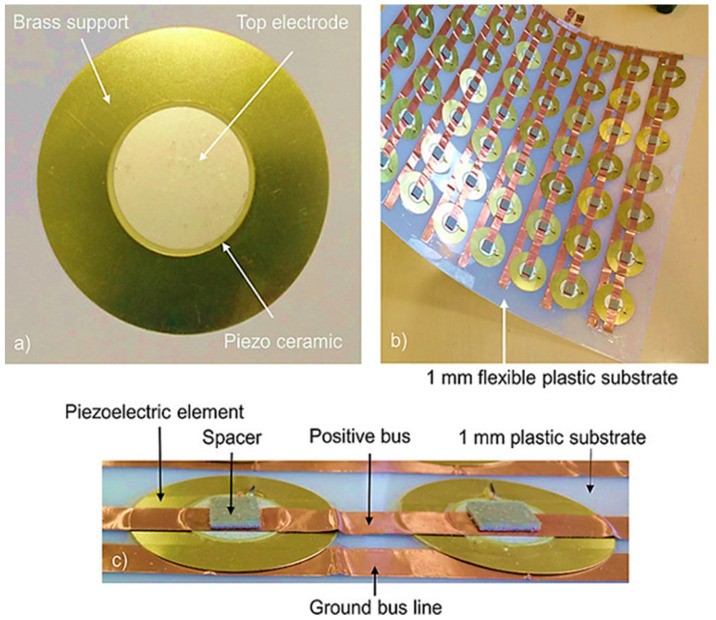
(**a**) Structure of the energy-harvesting tile active layer using commercially available piezoelectric materials (**b**) matrix of 8X7 elements on flexible substrate (**c**) structural detail of the arrays. [109].

**Figure 26 sensors-23-04227-f026:**
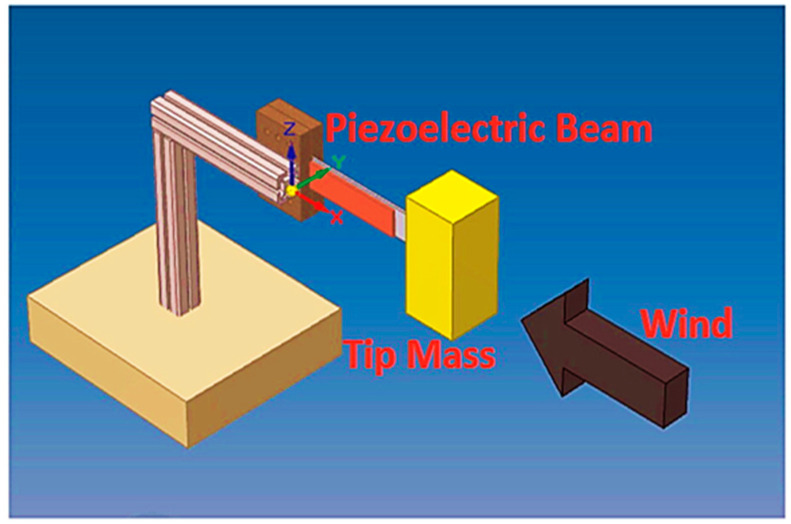
Mechanism of a galloping energy harvester [116].

**Figure 27 sensors-23-04227-f027:**
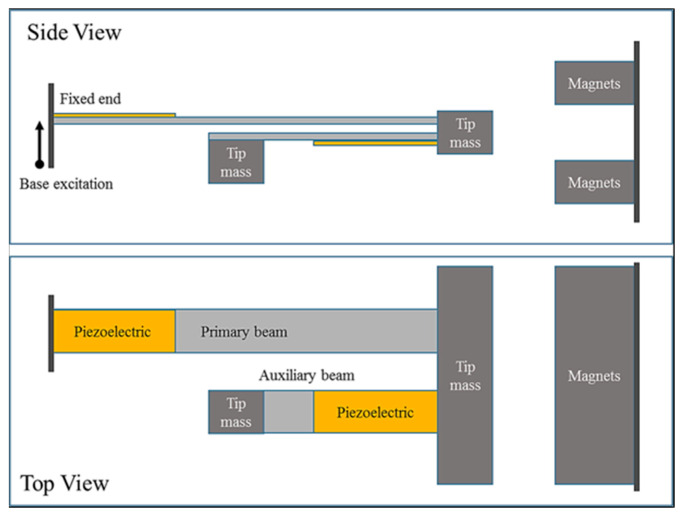
U-VPEH model design proposed by Sun et al. [121].

**Figure 28 sensors-23-04227-f028:**
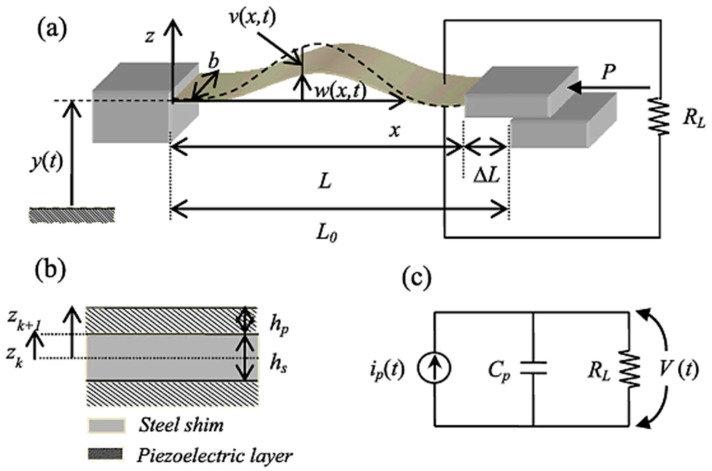
Buckled beam structure for vibration-based energy harvesting (**a**) schematic diagram of piezoelectric buckled bridge, (**b**) cross-section of the steel support and piezoelectric layer (**c**) electrical circuit of the system [128].

**Figure 29 sensors-23-04227-f029:**
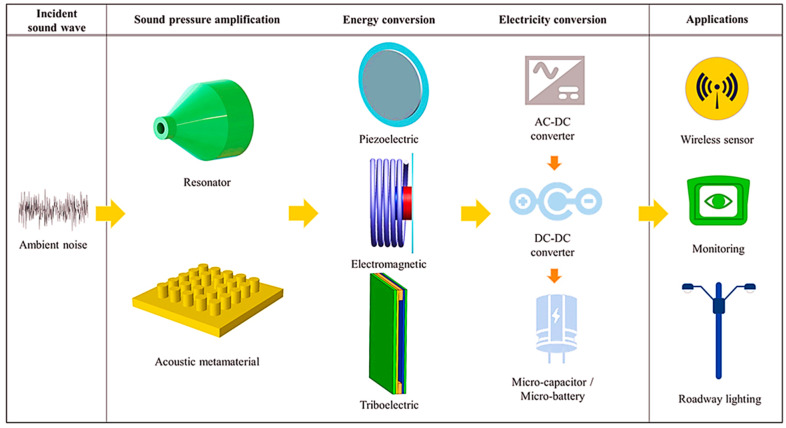
Sound energy harvesting mechanism [129].

**Figure 30 sensors-23-04227-f030:**
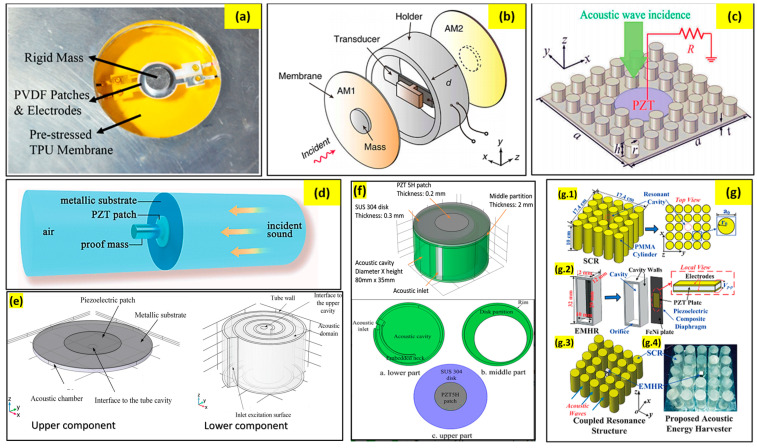
Acoustic energy harvesting structures. (**a**) Membrane-type sound absorber and energy harvester [130], (**b**) compact energy harvester with beam-based PZT [131], (**c**) planar acoustic metamaterial [132], (**d**) energy harvester using a metallic substrate and a proof mass [133], (**e**) 3D printed helix structure with PZT patch [134], (**f**) low-frequency acoustic energy harvester based on planar Helmholtz resonator [135], (**g**) acoustic energy harvesting using coupled sonic crystal and Helmholtz resonator: (**g.1**) structure diagram of the Sonic crystal resonator structure, (**g.2**) structure diagram of the electromechanical Helmholtz resonator structure, (**g.3**) structural diagram of the coupled resonance structure, (**g.4**) photograph of the coupled resonance structure [136].

**Figure 31 sensors-23-04227-f031:**
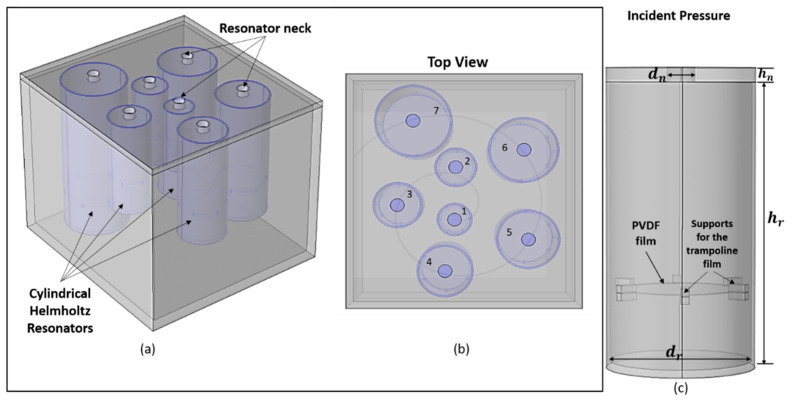
Spiral Helmholtz resonator-type acoustic energy harvester (**a**) entire block, (**b**) top view of the resonator block, numbers 1–7 representing the multiple Helmholtz resonators used inside the block with varying diameter, (**c**) side view of a single Helmholtz resonator [137].

**Figure 32 sensors-23-04227-f032:**
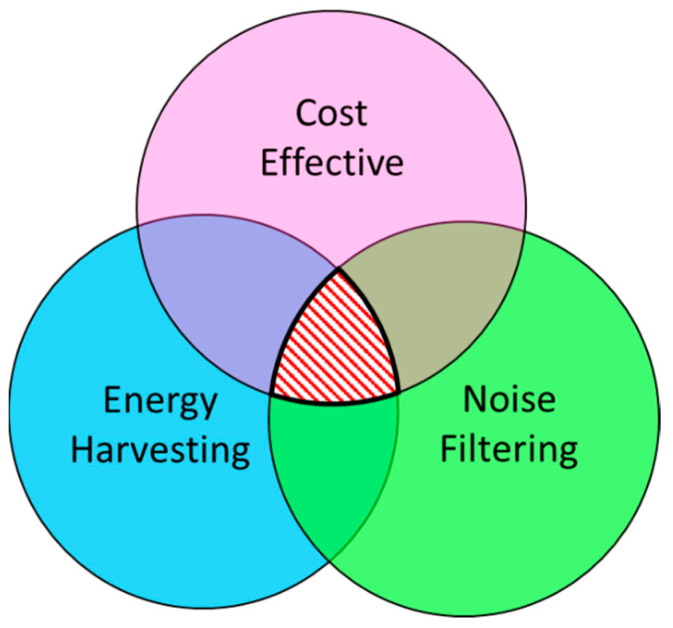
Selection criteria of the best noise barrier.

**Table 2 sensors-23-04227-t002:** Summary of passive control mechanisms with noise reduction capability.

Helmholtz Resonator			
Author		Model Description	Noise Reduction
Fahy and Schofield [27]		A cylindrical resonator with a cavity and neck placed inside a room is exposed to a various range of frequencies.	11.6 dB
Neise and Koopmann [30]		Replaced the scroll cutoff with a quarter-wavelength resonator. It was tuned by changing the length via a movable plug	29 dB
Franco et al. [31]		reduction of low-frequency cabin noise by tuning an acoustical resonator using the luggage compartment.	1.4 dB
Laudien et al. [37]		Reduction in helicopter cabin noise using honeycomb bulkhead and various measures to optimize transmission loss in a window, sealing, and frame.	17 dBA
Zhao et al. [38]		Rainbow trapping of acoustic waves using a hollow spiral tube and 40 Helmholtz resonators attached to it.	
Isozaki et al. [39]		Planar acoustic notch filter with multiple spherical Helmholtz resonators placed in the vertices of a polygon.	
Selamet and Lee [42]		Studied a concentric Helmholtz resonator with an extended neck.	~40 dB
Yang et al. [43]		Effects of different neck materials on the sound absorption capability of a Helmholtz resonator.	~0.75 (Sound absorption coefficient)
Tadeu and Mateus [91]		Experimentally validated the sound insulation capability of glazed openings.	30 dB (R_w_)
**Sound Absorbing Material**			
Yang et al. [45]		Experimentally verified the theoretical study of the membrane-type acoustic metamaterial in the 100–1000 Hz range.	0.01% (Transmission)
Yang et al. [46]		Metamaterial with two coupled membranes which has double negativity.	13.9 dB (STL)
Ang et al. [47]		Design and verification of meta panel made of hollow plexiglass tubes for low-frequency noise control.	
Cai et al. [48]		Coiling up quarter wavelength sound absorbing tubes in a coplanar matrix to form a sound absorbing panel.	100% (Absorption)
Cavalieri et al. [49]		Periodically coupled quarter wavelength and Helmholtz resonators to produce large insertion loss.	16.8 dB (Insertion Loss)
Kumar et al. [50]		Dual hexagonal resonators connected by a common neck for low-frequency noise absorption in aircraft.	TL: 58 dB Absorption: 48%
Li and Assouar [52]		A coiled coplanar air chamber and perforated plate used to construct a low-frequency perfect sound-absorbing metasurface.	100% (Absorption)
Zhu et al. [51]		Used the Schroeder diffuser design to develop a sound-diffusing acoustic metasurface.	
Zhang et al. [54]		Labyrinth acoustic metamaterial capable of perfectly absorbing low-frequency airborne sound.	100 % (Absorption)
Huang et al. [53]		Perfect sound absorber using coiled channel and embedded aperture.	100 % (Absorption)
**Acoustic Metamaterial**			
**Bragg Scattering**	Prasetiyo et al. [55]	Coiled-up air chamber to absorb low-frequency sub-wavelength sound.	80% (Absorption)
	Casadei et al. [58]	Phononic crystal plate with cylindrical stubs and L-shaped wave-guide with PZT discs.	
**Deaf Band Based**	Ao and Chan [69]	Locally resonant acoustic metamaterial capable of creating low-frequency bandgap which can be tuned for desired range; creates deaf bands.	
	Indaleeb et al. [70]	Targeted Dirac cone at a higher frequency validating orthogonal energy transport in a spiral pattern.	Dirac cone at 12.5 kHz and 18.512 kHz
	Indaleeb et al. [71]	Deaf band-based phononic crystals are modeled, and multiple occurrences of Dirac-like points are demonstrated.	Dirac cone at ~12.5 kHz and ~18.5 kHz
**Acoustic Quantum Hall Effect**	Wen et al. [72]	Uniform pseudo magnetic field in acoustics by adding de-formation in acoustic graphene.	
	Zhou et al. [73]	Membrane-type metamaterial developed with tunable topological properties to monitor the quantum valley–Hall effect.	Dirac cone at ~275 Hz
	Yves et al. [74]	Guiding sound waves at a lower scale than the operational wavelength and experimentally observing the quantum valley–Hall effect.	Dirac cone and bandgap above 355 Hz
**Topological Effect**	Fu and Kane [75]	The linear connection between super-conductors judged by a topological insulator to form a nonchiral 1D wire.	
	Zhang et al. [76]	Calculated pre-diction of topo-logical insulators with a single Dirac cone on the surface.	
	Chen et al. [77]	Investigating the surface state of Bi_2_Te_3_ to prove the existence of a single, nondegenerate Dirac cone in the surface state, also indicating a full energy gap for bulk states.	
	Lee and Iizuka [78]	Phononic metamaterial with C-shaped elements creates a topological bandgap due to the addition of resonance scattering.	Bandgap at ~5 kHz
	Xue et al. [79]	Higher order topological insulators using Kagome lattice structure with cylindrical resonator at each site.	
	Weiner et al. [80]	A 3D topological metamaterial displays the analog of pyrochlore lattice and shows topological bulk polarization.	
**Viscoelastic Damping System**			
Yu et al. [88]		Optimized formulation for viscoelastic damping of noise control in mid-frequency vibroacoustic systems.	Acoustic energy decrease: ~17.49 dB
Feng et al. [89]		Triple-layer adhesive glass capable of minimizing vibration in the existing glass window.	~66.7% reduction in amplitude
Valvano et al. [90]		Viscoelastic laminated panels capable of damping band frequencies are used for passive control of noise reduction.	

**Table 3 sensors-23-04227-t003:** Summary of passive control mechanisms with bandgaps.

Author		Model Description	Stop Band
**Bragg Scattering**	Ning et al. [56]	Low-frequency vibration control using a square array of circular holes and resonating mass.	~0.25 and 1 (Normalized frequency)
	Chen and Wang [57]	Triply periodic continuous acoustic metamaterials to absorb acoustic and elastic waves under harsh environments.	0.95 MHz and above
	Cai et al. [60]	Used FEM to calculate the band structure of Bragg scattering and locally resonant penta mode materials.	2 PBG below 400 Hz
	Wen et al. [61]	Acoustic metamaterial beam with periodically varying cross-section for enhanced bandgap properties.	
**Local Resonance**	Krushynska et al. [62]	Extensive study of LRAM analyzing bandgaps for in and out of plane modes.	Multiple bandgaps above 200 Hz
	Lim et al. [65]	Plate–pillar structure with multiple coatings of varying stiffness used for local resonance bandgaps and broadband vibration attenuation.	Bandgaps above 10 kHz
**Spring-mass damping system**	Peng and Pai [81]	Spring mass damper metamaterial plates create a stopband above the natural frequency of the subsystem.	500–534.5 Hz
	He et al. [82]	Carbon fiber-reinforced polymer and mass-spring dampers create metamaterials that create stopbands and act as a vibration absorber.	400–464.5 Hz
	Liu et al. [83]	Compound beam spring damper system to reduce noise from railway tracks.	
	Xiao et al. [84]	Study and verification of laminate acoustic metamaterial, which demonstrates multiple stopbands.	352–378 Hz and 442–465.5 Hz

**Table 4 sensors-23-04227-t004:** Summary of passive control mechanisms with energy harvesting and noise filtering.

Author	Model Description	Noise Reduction	Power Output
Wang et al. [40]	A hexagonal Helmholtz resonator with PVDF film inside is the cavity arranged in a honeycomb structure that is used to filter railway noise and harvest energy.	20.86 (Amplification ratio)	74.6 mV
Ahmed et al. [68]	Harvest energy and trap noise at lower sonic frequency using acoustoelastic metamaterial.	30.6 mPa (Pressure amplitude)	92.4 μW/cm^2^
Mir et al. [66]	Metamaterial wall with concrete frame proposed for industrial noise barrier.	57.7%	~1.2 mW
Ahmed and Banerjee [59]	Dynamic energy trapping inside a soft matrix of metamaterial for noise filtering and energy harvesting.		~10–90 μW

**Table 5 sensors-23-04227-t005:** Energy harvesting based on excitation sources.

Vibration Sources		
Author	Model Description	Power Output
Puscasu et al. [109]	Energy harvesting using pressure and vibration from steps in a busy corridor using piezoelectric membrane.	17.7 mJ
Lueke et al. [110]	Fixed-fixed folded spring-type vibration-based energy harvester for low-frequency energy harvesting.	690.5 nW
Liu et al. [111]	S-shaped PZT cantilever for very low frequency (<30 Hz) vibration and low acceleration (<0.4 g) energy harvester.	40 mV
Liu et al. [112]	Spiral-shaped PVDF cantilever for harvesting energy from a low frequency of around 20 Hz.	1.8 V
Bai et al. [113]	Spiral-shaped multi-modal vibration energy harvester with magnetoelectric transducers as tip mass for low-frequency energy harvesting.	~24 V
Zhao et al. [114]	Spiral-shaped thin elastic beam with a PZT layer and proof mass for low-frequency energy harvesting.	330.8 μW
Zorlu et al. [115]	Low-frequency MEMS energy harvester that generates energy from low displacement amplitude vibrations.	363 nW
Ewere and Wang [116]	Galloping piezoelectric energy harvester uses vibration from the wind with different tip bluff bodies to characterize its performance.	~9 mW
Yan et al. [117]	A galloping piezoelectric energy harvester with a general EM decoupled model, including the derivation and analysis of electrical damping corresponding to Hopfbirufication.	~4 mW
Wang et al. [119]	Tri-stable galloping piezoelectric energy harvester using non-linear magnetic force.	0.73 mW
Dash et al. [120]	Proposed non-linear EM distributed parameter model for GPEH and the effect of different order polynomials for aerodynamic force on dynamic behavior is investigated.	~14 mW
Sun et al. [121]	Linear and non-linear U-shaped vibration-based energy harvester with tip mass and magnets is investigated.	14.18 V
Hosseini et al. [122]	Comprehensive analysis of the relationship between the shape of the piezoelectric cantilever and voltage output and deducing a rule of thumb for calculation.	6.75 V
Jemai et al. [123]	Unimorph cantilever beam-type energy harvester analyzed under nonuniform vibration mode shapes and optimized the performance of the system.	~0.01 W
Zeng et al. [124]	Unimorph piezoelectric energy harvester with one through-width crack in the form of delamination to study the influence of the delamination on voltage and power output.	1.8 × 10^−19^
Tsujiura et al. [125]	Thin bimorph cantilever energy harvester capable of generating electric power using the self-excited vibration prompted by continuous airflow.	~53 μ
Yeo et al. [126]	Bimorph PCM energy harvester demonstrating high efficiency and power output from low-frequency mechanical vibration.	3.9 mWcm^−2^ g^2^
Alsaadi and Sheeraz [127]	A bimorph energy harvester that proves the effect of piezoelectric layer thickness on the energy output.	7 W
Cottone et al. [128]	Bistable oscillators exposed to non-linear vibration exhibit superior power generation in a wide resistance range.	10^−1^ μW
**Sound Sources**		
Li et al. [130]	Piezoelectric patch on either side of a thin membrane to harvest energy from the strain created in the membrane from sound waves.	~10 nW 15.3% (energy conversion efficiency)
Wang et al. [131]	Beam-based PZT transducer with two layers of acoustic metamaterial which increases the efficiency and power output by 4.2 times than a transducer without LAM.	72.6 mV
Qi et al. [132]	PZT patch attached to a defect in the AMM plate with an array of silicone rubber stubs on a thin aluminum plate which harvests energy from acoustic pressure.	1.3 V 0.54 μW/cm^3^ (power density)
Yuan et al. [133]	Helical acoustic resonator, which can be 3D printed and occupies less volume, is capable of harvesting energy from the piezoelectric patch bonded on the cap using sound pressure.	7.3 μW
Yuan et al. [134]	A metallic substrate with proof mass is designed to harvest energy from acoustic energy, which overcomes the drawbacks of the rubber film.	0.21 mW
Yang et al. [136]	Coupled acoustic resonance of sonic crystal and Helmholtz resonator to magnify acoustic pressure and harvest higher pressure.	429 μW
Yuan et al. [135]	Helmholtz resonator with tapered neck and PZT patch on the cover to harvest low-frequency acoustic energy.	64.4 μW
Mir et al. [137]	Helmholtz resonators arranged in a spiral pattern to block noise and harvest energy from the acoustic pressure.	0.7 μW

## Data Availability

All Data are available upon reasonable request made to the corresponding author.

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
