# Peer review of "Metamaterials for Acoustic Noise Filtering and Energy Harvesting"

_sensors, 2023, doi:10.3390/s23094227_

Round 1
Reviewer 1 Report
Review Report on “Metamaterials for Energy Harvesting & Acoustic Noise Filter- 2
ing” by Mir et al.
This study introduces the twenty-first century's Factory vision 4.0 that has brought with it a new set of challenges, including increased noise pollution from manufacturing, engineering, and transportation industries. To address this issue, researchers are exploring new engineered materials called Acoustic Metamaterials (AMM), which have unique properties that allow them to filter ambient noise while also harvesting energy from sound and vibration sources. In this review, we will explore the various AMM that have been used for noise filtering and highlight areas for further research.
In summary, this research article explores the potential of AMM for noise filtering and energy harvesting. The authors provide background information on the challenges posed by noise pollution and the potential benefits of AMM for addressing this issue. They then review various designs for AMM-based noise barriers and highlight the potential for energy harvesting from sound and vibration sources. The article concludes with recommendations for further research in this area.
While the article provides a comprehensive review of the potential of AMM for noise filtering and energy harvesting, there are some limitations to the research which may be addressed in the revised version. For example;
1) The authors focus primarily on the design of AMM-based noise barriers and do not explore other potential applications of these materials.
2) Additionally, the article does not provide a detailed analysis of the effectiveness of these barriers in real-world settings, which would be important for understanding their practical applications. However, the article does highlight some important areas for future research, including the development of more efficient AMM designs and the exploration of other applications for these materials.
Overall, this research article provides a valuable review of the potential of Acoustic Metamaterials for noise filtering and energy harvesting. While there are some limitations to the research, the article highlights the significant potential of these materials for addressing the challenges of noise pollution in the twenty-first century. The authors' recommendations for further research provide a useful guide for future exploration in this area.
Therefore, I would suggest revising the article thoroughly thereby improving the presentation considerably. For example:
1. The abstract does not describe the actual essence of the problem. Therefore, it should be modified so that it should provide a concise summary of the study's purpose, methods, results, and conclusions. It should be clear and informative, and briefly describe the context or background of the study, the methods used, the main findings, and the implications of the study's results. I also see grammatical errors in the abstract which must be carefully addressed.
2. In the introduction section, the literature is adequately addressed in most of the parts. However, I would suggest referring more relevant, recent and scientific studies about noise control in the 2nd paragraph of the introduction, for instance, “A note on noise source localization, Vibration and Control, 22(7): (2016), 1889-1894”, Scattering characteristics of non-planar trifurcated waveguides, Meccanica, (2020), 55, 971-988.” and Analytical prediction of the breakout noise from a rectangular cavity with one compliant wall. Journal of Acoustical Society of America 124: 2952–296 etc.
3. I hardly saw the mathematical aspect of the study except that this scientific study is performed theoretically. I would like to see authors’ comments in this context.
4. The conclusion of the article is not up to the mark and does not conclude the study in its actual essence. The conclusion should provide a concise summary of the study's main findings and offer a clear statement of the study's implications and contributions to the underlying research. It should summarize the key results, address the research question, discuss any limitations of the study, offer implications and recommendations, and end with a strong remark. The conclusion should be concise, focused, and tightly linked to the study's purpose and findings, and should avoid introducing new information or arguments.
5. The article should be proofread thoroughly as it contains a lot of typos/grammatical errors which should be carefully addressed.

Reviewer 2 Report
This paper presents a detailed review of the metamaterials developed so far that can be used for active noise filtering and energy harvesting. It is a detailed review paper stating state of art in the field of noise filtering and energy harvesting. It gives a brief summary of all methods used and very comprehensive.
Reviewer 3 Report
This is a nice review research on the metamaterials for energy harvesting & acoustic noise filtering. However, some important latest references are missing. It is recommended that more references in the last three year should be added and discussed.
Reviewer 4 Report
The manuscript is an excellent and detailed review of the metamaterials developed, which are used for active noise filtering and energy harvesting.
In the reviewer's opinion, the content of the review manuscript is suitable to be accepted for publication in the Journal. There are two important point to consider in the final version of the article:
1.- The results shown in the different manuscript figures, should be explained in more detailed way. For instance, it's the case of results of results presented in Fig. 2.10, 2.12, 2.19, etc.
2.- Considering that some figures come from their original papers, notice that is important to have enough quality in these figures in order to read the text, labels, etc. It's the case of Figs. 2.19, 2.22 or 3.7.
Reviewer 5 Report
This work presents a comprehensive review of noise-filtering techniques with a specific focus on acoustic metamaterials. The review paper has been divided into two major sections, including noise control technologies and energy harvesters, each having several subsections. Regarding the noise control devices, the authors mentioned both active and passive systems and described the pros and cons of each available technology with sufficient detail. More interestingly, the present work summarized the past research on active and passive noise control systems in tables 2.1 and 2.2, respectively. These tables can help the readers to access their necessary information much faster. The same procedure is applied to the case of energy harvesting systems. These systems have been categorized into two sub-categories based on their source, e.g., vibration and sound sources. In each scenario, the authors describe the physics behind each phenomenon and then present the possible applications. Eventually, some recommendations have been proposed for future multifunctional designs of noise-control systems.
The introduction provides sufficient background knowledge of the topic, which can be useful even for non-expert readers. Overall, the paper is very well-organized, containing a complete literature review that can provide new insights into the design of noise control devices and energy harvesters. After addressing the following comments, I would recommend the paper for publication in Sensors.
Major Comment
· Did authors get the necessary permissions to reuse the figures from other references in their study? If yes, they should explicitly declare it.
· The authors must pay particular attention to the citations and bibliography in these types of review papers. It seems that the authors modified some citations inside the text and forgot to update the bibliography. On the other hand, the references in the tables match the reference list and numbering. I checked the citations and found that the citations are wrongly numbered from line 208 up to the end of the paper. Below, you can find some examples.
1. Line 208, Martin et al. is ref no. 5, not 7.
2. Line 212, Kestell et al. is ref 6.
3. Line 231, Fuller and Jones is ref 10.
4. Line 240, Fuller et al. is ref 15.
5. Line 273, Fahy and Schofield is ref 21.
6. And so on …
Some papers are disappeared from the reference list, for instance:
1. Line 214, Samara-singhe et al. [13] is not listed. Ref 13 is something else.
2. Line 438, Muhammad and Lim [63] are not listed, while ref no. 63 is another study.
These are just some errors in the text; therefore, the authors are encouraged to review the references with special care and attention.
· Regarding the local resonance mechanism (section 2.1.2.3.2) appeared in which leads to the generation of low-frequency bandgaps, the authors are encouraged to cite the pioneering work by the group of Prof. Ping Sheng, Liu et al. (2000)*1 where the local resonance mechanism appears in a rigid core attached to host medium via soft material.
Moreover, when authors discuss the observance of bandgaps in thin elastic plates within lines 438-440 of the manuscript, the authors may cite the famous experimental work by Colombi et al.(2016)*2 in which forest trees are considered as working as natural locally resonant metasurfaces, instead of Muhammad and Lim which is not defined in the references. The authors can even use one of the schematics of these papers instead of Fig 2.7b.
In addition to these works which consider the local resonance of metasurfaces as an ultra-thin resonant layer, the authors may consider citing the work by Zeighami et al. (2021)*3 in which they consider a thick resonant layer composed of concentric local resonators. This configuration leads to the design of barriers for both Surface Acoustic Waves devices and seismic isolation systems.
· In the spring-mass-damping systems (Section 2.1.2.4), the authors could also refer to the work by Colquitt et al.(2017)*4 in which the interaction between surface Rayleigh waves and sub-wavelength resonators (which can be conceived as mass-spring systems) analytically and numerically. The experimental study of such a system has been done in a study by Palermo et al. (2016)*5.
· Besides the local resonance and Bragg scattering mechanism, the authors can indicate the other possible method, the inertial amplification mechanism, initially proposed by Yilmaz et al. (2007)*6.
Minor Comments
1. The energy harvesting concept has been described in lines 41 and then 134. It would be better to combine them.
2. Line 604 is table 2.2, not 2.1.
Typos
1. Line 136, necessary to *the* ensure.
2. Line 201, the late 80s *where* researchers.
3. Line 231, *t*he initial fundamental.
4. Line 232, achieved *a* cabin noise reduction.
5. Line 268, hollow container*s*.
6. Line 370, properties that *allow* them.
7. Line 388: Bragg scattering can be *observedIt* happens at the heart of the phononic crystal.
8. Line 428, for the *filtration* of sound.
9. Line 627, *could* be seen
10. Line 951, to *a* real-world scenario.
Proposed additional references
Here is a list of possible papers that can contribute to enhancing the quality of this review paper. The authors may accept or reject these suggestions.
*1 Liu, Z., Zhang, X., Mao, Y., Zhu, Y.Y., Yang, Z., Chan, C.T., Sheng, P.,2000. Locally resonant sonic materials. Science 289, 1734–1736
*2 Colombi, A. et al. Forests as a natural seismic metamaterial: Rayleigh wave bandgaps induced by local resonances 2016. Sci. Rep. 6, 19238; doi: 10.1038/srep19238.
*3 Zeighami, F., Palermo, A., Marzani, A., 2021. Rayleigh waves in locally resonant metamaterials. International Journal of Mechanical Sciences 195, 106250. doi:https://doi.org/10.1016/j.ijmecsci.2020.106250.
*4 Colquitt, D., Colombi, A., Craster, R., Roux, P., Guenneau, S., 2017. Seismic metasurfaces: Subwavelength resonators and Rayleigh wave interaction. Journal of the Mechanics and Physics of Solids 99, 379–393. doi:https://doi.org/10.1016/j.jmps.2016.12.00
*5 Palermo, A. et al, 2016. Engineered metabarrier as shield from seismic surface waves. Sci. Rep. 6, 39356; doi: 10.1038/srep39356.
*6 C Yilmaz, GM Hulbert, N Kikuchi, 2007, Phononic band gaps induced by inertial amplification in periodic media, Physical Review B 76, 054309. Doi: doi.org/10.1103/PhysRevB.76.054309.
Round 2
Reviewer 5 Report
The authors revised the manuscript with full consideration of my previous comments. Therefore, I would suggest the publication of this review paper in its current format.